# Progressive Inference-Time Annealing of Diffusion Models for Sampling from Boltzmann Densities

**Tara Akhound-Sadegh**[1,2][*] **Jungyoon Lee**[3,2][*] **Avishek Joey Bose**[4,2], **Valentin De Bortoli**[5],
**Arnaud Doucet**[5], **Michael M. Bronstein**[4,6], **Dominique Beaini**[7,3,2], **Siamak Ravanbakhsh**[1,2],
**Kirill Neklyudov**[3,2,8][†], **Alexander Tong**[6,3,2][†]

[1]McGill University, [2]Mila – Quebec AI Institute, [3]Université de Montréal, [4]University of Oxford,
[5]Google DeepMind, [6]AITHYRA, [7]Valence Labs, [8]Institut Courtois

## Abstract

Sampling efficiently from a target unnormalized probability density remains a core challenge, with relevance across countless high-impact scientific applications. A promising approach towards this challenge is the design of amortized samplers that borrow key ideas, such as probability path design, from state-of-the-art generative diffusion models. However, all existing diffusion-based samplers remain unable to draw samples from distributions at the scale of even simple molecular systems. In this paper, we propose PROGRESSIVE INFERENCE-TIME ANNEALING (PITA) a novel framework to learn diffusion-based samplers that combines two complementary interpolation techniques: I.) Annealing of the Boltzmann distribution and II.) Diffusion smoothing. PITA trains a sequence of diffusion models from high to low temperatures by sequentially training each model at progressively lower temperatures, leveraging engineered easy access to samples of the temperature-annealed target density. In the subsequent step, PITA enables simulating the trained diffusion model to *procure training samples at a lower temperature* for the next diffusion model through inference-time annealing using a novel Feynman-Kac PDE combined with Sequential Monte Carlo. Empirically, PITA enables, for the first time, equilibrium sampling of $N$-body particle systems, Alanine Dipeptide, and Tripeptide in Cartesian coordinates with dramatically fewer energy function evaluations. Code available at: https://github.com/taraak/pita.

## 1 Introduction

The problem of sampling from an unnormalized target probability distribution arises in numerous areas of natural sciences, including computational biology, chemistry, physics, and materials science (Frenkel and Smit, 2023; Liu, 2001; Ohno et al., 2018; Stoltz et al., 2010; Noé et al., 2019). In many of these high-impact scientific settings, the problem's complexity stems from operating in molecular systems where the unnormalized target (Boltzmann) distribution at a low temperature of interest is governed by a highly non-convex and non-smooth energy function, under which there is limited to no available data (Hénin et al., 2022). As a result, the sampling problem necessitates solving an equally hard exploration problem: finding the modes—in correct proportion—of the target distribution.

To address the general sampling problem, extensive research has been dedicated to Markov chain Monte Carlo methods (MCMC), Sequential Monte Carlo (SMC), and, particularly in physical systems, Molecular Dynamics (MD) (Leimkuhler and Matthews, 2015). To enhance scalability, Monte Carlo approaches often employ an interpolating sequence of probability distributions that

---

[*]Equal contribution [†]Equal advising. Correspondence to `tara.akhoundsadegh@mila.quebec`, `k.necludov@gmail.com`, `atong@aithyra.at`

39th Conference on Neural Information Processing Systems (NeurIPS 2025).

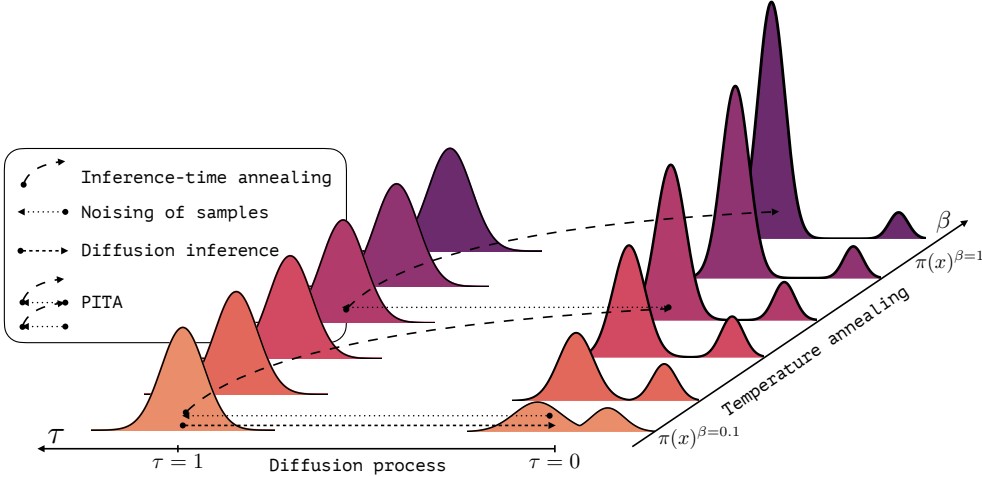

Figure 1: Illustration of the proposed PITA framework combining two complementary processes: temperature annealing of the target Boltzmann density and the diffusion process applied to the collected samples. Annealed inference allows for decreasing the temperature (increasing $\beta$) of a trained diffusion model, thus generating samples from the annealed target. These samples can be reused for training a lower-temperature diffusion model.

transitions from an easily sampled reference distribution to the desired target distribution via annealing/tempering strategies. This powerful concept underlies methods such as parallel tempering (Swendsen and Wang, 1986), Annealed Importance Sampling (Jarzynski, 1997; Neal, 2001), and SMC samplers (Del Moral et al., 2006). MD, conversely, involves integrating equations of motion using finely discretized time steps. Despite their effectiveness, both classes of methods possess inherent limitations that complicate their application to practical systems of interest: annealing modifies the masses of distribution modes depending on their widths (a phenomenon often referred to as mass teleportation (Woodard et al., 2009)), while MD requires computationally expensive time discretization on the order of femtoseconds to simulate millisecond-scale phenomena.

Diffusion-based samplers are an alternative and emergent class of sampling techniques (Zhang and Chen, 2022; Vargas et al., 2023; Akhound-Sadegh et al., 2024; Berner et al., 2024; Blessing et al., 2024; Havens et al., 2025) exploiting modern developments in generative modeling. They sample complex multi-modal distributions by leveraging a prescribed interpolating probability path. However, instead of relying on annealing, these samplers utilize a noising mechanism which theoretically enjoys favorable mixing properties compared to annealing (Chen et al., 2023).

Diffusion-based samplers, despite their appeal, have not yet proven effective for even small molecular systems in Cartesian coordinates. This is primarily because of the absence of training data to accurately approximate the logarithmic gradients of the marginal densities, i.e. the Stein scores – a challenge distinct from generative modeling settings. Additionally, standard training objectives, such as reverse Kullback–Leibler, are mode dropping and often yield too high-variance score estimates for stable training (Blessing et al., 2024). Crucially, current diffusion-based samplers require too many energy function evaluations for training. Indeed, when normalized by the number of energy evaluations, carefully tuned MCMC methods with parallel tempering are empirically competitive with, if not superior to, state-of-the-art diffusion-based samplers (He et al., 2025).

**Main Contributions**. In this paper, we introduce PROGRESSIVE INFERENCE-TIME ANNEALING (PITA), a novel framework for training diffusion models to sample from Boltzmann distributions. PITA leverages two complementary interpolation techniques to significantly enhance training scalability: temperature annealing (increasing the system's temperature) and interpolation along a conventional diffusion model's probability path. This combination is motivated by a learning framework designed to exploit their distinct advantages: temperature annealing mixes modes by lowering high-energy barriers, while diffusion paths avoid mass teleportation.

Annealing the target distribution transforms the challenging sampling problem into an easier one by removing high-energy barriers and flattening it. This crucial step enables the cheap collection of an initial high-temperature dataset via classical MCMC, which in turn facilitates the efficient training of an initial diffusion model. Subsequently, we define a novel Feynman-Kac PDE that, when combined with SMC-based resampling, allows us to simulate the trained diffusion model's inference process to produce asymptotically unbiased samples at a lower temperature. This effectively allows us to train the next diffusion model, enabling the progressive and stable training of a sequence of diffusion models until the target distribution is reached, as illustrated in Fig. 1.

We test the empirical performance of PITA on standard $N$-body particle systems and short peptides in Alanine Dipeptide and tripeptides. Empirically PITA not only achieves state-of-the-art performance in all these benchmarks but is the first diffusion-based sampler that scales to our considered peptides in their native Cartesian coordinates. More importantly, we demonstrate that progressing down our designed ladder of diffusion models leads to significantly lower energy evaluations compared to MD, which is a step towards realizing the promise of amortized samplers for accelerating equilibrium sampling.

## 2 Background

### 2.1 Diffusion models

A diffusion process defines an interpolating path between an easy-to-sample reference density, such as a multi-variate Normal, and a desired target distribution $\pi(x)$. When samples from the target distribution are available, it is possible to generate samples from intermediate marginals of the diffusion process $p_\tau(x)$ through the following Gaussian convolution:

$$p_\tau(x) = \left(\pi * \mathcal{N}(\alpha_\tau y, \sigma_\tau^2 \mathbb{1})\right)(x) = \int dy\, \mathcal{N}(x \,|\, \alpha_\tau y, \sigma_\tau^2 \mathbb{1})\pi(y). \tag{1}$$

As a result, this means that samples from $p_\tau(x)$ can be generated as $x_\tau = \alpha_\tau y + \sigma_\tau \varepsilon$, where $y \sim \pi(y)$ and $\varepsilon \sim \mathcal{N}(0, \mathbb{1})$. Selecting specific schedules for $\alpha_\tau$ and $\sigma_\tau$ one can ensure the following boundary conditions. For $\tau = 0$, $\alpha_\tau = 1$, $\sigma_\tau = 0$ and $p_{\tau=0}(x) = \pi(x)$, i.e. the marginal matches the target distribution. For $\tau = 1$, $\alpha_\tau = 0$, $\sigma_\tau = 1$ and $p_{\tau=1}(x) = \mathcal{N}(0, \mathbb{1})$, i.e. the marginal matches the standard multivariate normal distribution.

Importantly, despite the simplicity of sample generation, the evaluation of density $p_\tau(x)$ is not straightforward, and one has to use deep learning models to approximate either scores $\nabla \log p_\tau(x)$ or marginal densities $p_\tau(x)$. Furthermore, the model density $p_\tau(x)$ or scores $\nabla \log p_\tau(x)$ can be used to generate new samples from $\pi(x)$ using the reverse-time SDE. In particular, the marginals introduced in Eq. (1) describe the marginal densities of the forward-time (Ornstein–Uhlenbeck) SDE

$$dx_\tau = \underbrace{\frac{\partial \log \alpha_\tau}{\partial \tau} x_\tau}_{:=a_{1-\tau} x_\tau} d\tau + \underbrace{\sqrt{2\sigma_\tau^2 \frac{\partial}{\partial \tau} \log \frac{\sigma_\tau}{\alpha_\tau}}}_{:=\zeta_{1-\tau}} d\overline{W}_\tau, \quad x_{\tau=0} \sim \pi(x), \tag{2}$$

where $\overline{W}_\tau$ is the standard Wiener process, and marginals follow the Fokker-Planck PDE. After inverting the time variable in this PDE, i.e. $t = 1 - \tau$, the time-evolution of marginals $p_t(x)$ is

$$\frac{\partial p_t(x)}{\partial t} = \langle \nabla, p_t(x)(a_t x) \rangle - \frac{\zeta_t^2}{2} \Delta p_t(x) = -\left\langle \nabla, p_t(x)\left(-a_t x + \frac{\zeta_t^2}{2} \nabla \log p_t(x)\right) \right\rangle, \tag{3}$$

which shows that one can sample from the marginals $\{p_t(x)\}_{t \in [0,1]}$ via simulating the following SDE

$$dx_t = \left(-a_t x + \frac{\zeta_t^2}{2}(1 + \xi_t)\nabla \log p_t(x_t)\right)dt + \zeta_t \sqrt{\xi_t}\, dW_t. \tag{4}$$

While the marginals are correct for any $\xi_t > 0$, there are two important special cases: for $\xi_t \equiv 1$, the equation becomes the reverse-SDE with the same path-measure as Eq. (2), and, for $\xi_t \equiv 0$, the SDE becomes an ODE. In practice, this SDE is simulated using either the density model $\exp(-U_t(x; \eta)) \propto p_t(x)$ (Du et al., 2023) or the score model $s_t(x; \theta) \approx \nabla \log p_t(x)$ (Song et al., 2021).

## 2.2 Annealing

Annealing defines a family of "simpler" problems when we have access to the unnormalized density by interpolating or scaling the target log-density (negative energy). Formally speaking, given the unnormalized density $\pi(x)$, the annealed density is defined as

$$\pi^\beta(x) := \frac{\pi(x)^\beta}{Z_\beta}, \quad Z_\beta = \int dx \, \pi(x)^\beta, \tag{5}$$

where $\beta$ is the inverse temperature parameter (i.e., $\beta = 1/T$) controlling the smoothness of the target density. Thus, for high temperature $T > 1$ (hence, $\beta < 1$), the target density becomes smoother and easier to explore via MCMC algorithms. Importantly, getting the unnormalized density for $\pi^\beta(x)$ can be simply achieved by raising the unnormalized density $\pi(x)$ to the power $\beta$.

## 2.3 Feynman-Kac Formula

The Feynman-Kac Partial Differential Equation (PDE) is a generalization of the Fokker-Planck PDE and is defined as follows

$$\frac{\partial p_t(x)}{\partial t} = -\langle \nabla, p_t(x) v_t(x) \rangle + \frac{\zeta_t^2}{2} \Delta p_t(x) + p_t(x) \big( g_t(x) - \mathbb{E}_{p_t(x)} g_t(x) \big), \tag{6}$$

where the first term corresponds to the probability mass transport along the vector field $v_t(x)$, the second term corresponds to the stochastic moves of samples according to the Wiener process $W_t$, and the last term is responsible for reweighting the samples according to a coordinate dependent weighting function $g_t(x)$. For any test-function $\varphi(x)$, the Feynman-Kac formula relates its expected value to the expectation over the SDE trajectories $x_t$, i.e.

$$\mathbb{E}_{p_T(x)}[\varphi(x)] = \frac{1}{Z_T} \mathbb{E}\Big[ e^{\int_0^T dt \, g_t(x_t)} \varphi(x_T) \Big], \quad \text{where } dx_t = v_t(x_t) dt + \zeta_t dW_t, \; x_0 \sim p_0(x), \tag{7}$$

and $Z_T$ is a normalization constant independent of $x$. In practice, the exponential term is computed as a "weight" $w_t$ of the corresponding sample $x_t$ and can be integrated in parallel with the simulation,

$$dx_t = v_t(x_t) dt + \zeta_t dW_t, \quad d \log w_t = g_t(x_t) dt, \quad \text{initialized as } x_0 \sim p_0(x), \; \log w_0 = 0. \tag{8}$$

Finally, one can estimate the normalization constant $Z_T$ by considering $\varphi(x) \equiv 1$ in Eq. (7) and get the biased but consistent Self-Normalized Importance Sampling (SNIS) estimator (Liu, 2001), i.e.

$$\frac{1}{Z_T} \mathbb{E}\Big[ e^{\int_0^T dt \, g_t(x_t)} \varphi(x_T) \Big] = \frac{\mathbb{E} e^{\int_0^T dt \, g_t(x_t)} \varphi(x_T)}{\mathbb{E} e^{\int_0^T dt \, g_t(x_t)}} \approx \sum_{i=1}^n \frac{w_T^i}{\sum_{j=1}^n w_T^j} \varphi(x_T^i), \tag{9}$$

where $(x_T^i, w_T^i)$ are the samples from the simulation of the SDE in Eq. (8).

# 3 Progressive Inference-Time Annealing

In this section, we combine diffusion and annealing processes into an efficient learning algorithm for sampling from the target density $\pi(x)$. To design this method, we build on the fact that diffusion and annealing are complementary ways to simplify or "smoothen" the target distribution (see Fig. 1). Namely, for the high-temperature version of the target distribution $\pi^{\beta_i}(x)$, we assume having samples from $\pi^{\beta_i}(x)$ and learn the density model of the marginals defined by the diffusion process (see Section 3.2). For instance, this can be done by running MCMC chains that face little challenge mixing in high temperatures. For the given density model of the diffusion process, we perform annealing of all the marginals and generate samples from the lower temperature target $\pi^{\beta_{i+1}}(x), \beta_{i+1} < \beta_i$ (see Section 3.1). We detail every step of our method in the following subsections.

## 3.1 Inference-Time Annealing

In this section, we discuss the inference-time annealing process, which allows us to modify the trained diffusion model to generate samples from the lower temperature target density. Namely, for a diffusion process with marginals $p_t(x)$ and the end-point $p_{t=1}(x) = \pi^{\beta_i}(x)$, we assume having two models: a score model $s_t(x; \theta) \approx \nabla \log p_t(x)$ and an energy-based model $U_t(x; \eta) \approx -\log p_t(x) + \text{const}$

with parameters $\theta$ and $\eta$ respectively. Given the score and the energy models trained to sample from a higher temperature density $\pi^{\beta_i}(x)$, we define a new sequence of marginals that correspond to the Boltzmann density of the energy model but with a lower temperature

$$q_t(x) \propto \exp(-\beta_{i+1}/\beta_i U_t(x;\eta)), \ q_{t=1}(x) \propto \exp(-U_{t=1}(x;\eta))^{\beta_{i+1}/\beta_i} \approx (\pi(x)^{\beta_i})^{\beta_{i+1}/\beta_i} . \quad (10)$$

The following proposition derives the Feynman-Kac PDE that describes the time-evolution of the marginals $q_t(x)$ and allows for importance sampling via the Feynman-Kac formula.

**Proposition 1.** [Inference-time Annealing] *Annealed density of the energy-based model $q_t(x) \propto \exp(-\gamma U_t(x;\eta))$ matches the marginal densities of the following SDE*

$$dx_t = \left( -a_t x_t + \frac{\zeta_t^2}{2}(s_t(x_t) - \gamma \xi_t \nabla U_t(x_t;\eta)) \right) dt + \zeta_t \sqrt{\xi_t} dW_t , \ x_0 \sim q_{t=0}(x) \quad (11)$$

$$d \log w_t = \left[ \frac{\zeta_t^2}{2} \langle \nabla, s_t(x_t) \rangle - \gamma \left\langle \nabla U_t(x_t;\eta), -a_t x_t + \frac{\zeta_t^2}{2} s_t(x_t) \right\rangle - \gamma \frac{\partial U_t(x_t;\eta)}{\partial t} \right] dt , \quad (12)$$

*where $s_t(x)$ is any vector field, $a_t, \zeta_t, \xi_t$ are analogous to parameters from Eq. (4), and the sample weights $w_t$ correspond to the SNIS estimator of the Feynman-Kac formula in Eq. (9).*

See Appendix A.1 for the proof. Intuitively, this result defines an importance sampling scheme, where Eq. (11) generates samples from the proposal distribution and Eq. (12) integrates the density ratio between the sampled proposal and the target density $q_t(x)$. Different choices of the vector field $s_t(x)$ and the noise schedule $\xi_t$ yield different proposal distributions. Theoretically, one can choose different parameters $a_t, \zeta_t$ as well, but below we argue for setting them according to Eq. (2).

The dynamics in Proposition 1 are not unique, and there exists a continuous family of PDEs that follow the marginals $q_t(x) \propto \exp(-\gamma U_t(x;\eta))$. We motivate this dynamics as minimizing the variance of the weights for the case when there is no annealing ($\gamma = 1$). Indeed, if the trained EBMs and score models approximate the diffusion process perfectly, then, for $\gamma = 1$, the weights become constant, so SNIS equally weights all the samples; thus, eliminating the need to resample at all. We formalize this result in the following proposition (see Appendix A.1 for the proof).

**Proposition 2.** [Convergence to Diffusion] *For $\gamma = 1$ and perfect models $s_t(x) = -\nabla U_t(x;\eta) = \nabla \log p_t(x)$, the variance of the weights in Proposition 1 becomes zero.*

In the case of unbounded support of the target distribution, e.g. $\text{supp}(\pi(x)) = \mathbb{R}^d$, increasing the temperature might cause numerical instabilities. Indeed, $\pi(x)^{\beta=0} \propto \text{Uniform}(\mathbb{R}^d)$ is not normalizable. To avoid this potential issue, in Appendix A.2, we consider geometric averaging between some simple prior and the target densities, e.g. $\mathcal{N}(0,\mathbb{1})^{(1-\beta)} \pi(x)^\beta$.

Integrating the dynamics from Propositions 1 and 3 we generate a set of weighted samples $\{(x_{t=1}^k, w_{t=1}^k)\}_{k=1}^K$ that converge to the samples from $q_{t=1}(x)$ when $K \to \infty$. In practice, this density is defined as the Boltzmann distribution of the corresponding energy model, i.e. $q_{t=1}(x) \propto \exp(-\beta_{i+1}/\beta_i \cdot U_{t=1}(x;\eta))$, which approximates $\pi^{\beta_{i+1}}(x)$, but does not necessarily match it exactly. We discuss several possible ways to bridge this gap between the density model and the target density in Appendix B.

## 3.2 Training using PITA

The proposed algorithm consists of interleaving the inference-time annealing (described in the previous two subsections) and model training on the newly generated data from the annealed target distribution, which we describe here. Throughout this stage we assume availability of samples from $\pi^{\beta_{i+1}}$, which were previously generated at the sampling stage[2].

For the target distribution $\pi^{\beta_{i+1}}(x)$, we define the diffusion process with the marginals $p_t(x)$ obtained as a convolution of the samples from the target $x \sim \pi^{\beta_{i+1}}(x)$ with the Gaussian $\mathcal{N}(\alpha_\tau x, \sigma_\tau^2 \mathbb{1})$. To

---

[2]For the very first iteration of our algorithm, we assume that there exist such $\beta$ that samples from $\pi^\beta$ can be simply collected by conventional Monte Carlo algorithms.

---

**Algorithm 1** Training for single temperature $1/\beta_{i+1}$

---

**Require:** samples $x$ from $\pi^{\beta_{i+1}}$.
  **for** training iterations **do**
      sample $\ln(\sigma_{1-t}) \sim \mathcal{N}(P_{mean}, P_{std}^2)$
      add noise $x_t \sim \mathcal{N}(x_t \,|\, \alpha_{1-t}x, \sigma_{1-t}^2 \mathbb{1})$
      Denoising Score Matching$(\theta) = \nabla_\theta \, \mathbb{E}_{t,x_t,x} \lambda(t)\|x - D_t(x_t;\theta)\|^2$
      Target Score Matching$(\theta) = \nabla_\theta \, \mathbb{E}_{t,x_t,x} \left[ \left\| \sigma_t^2 \nabla_x \log \pi(x) + x - D_t(x_t;\theta) \right\|^2 \cdot \mathbb{I}(t \geq t_{\text{thresh}}) \right]$
      EBM Distillation$(\eta) = \nabla_\eta \, \mathbb{E}_{t,x_t,x} \lambda(t) \left\| \sigma_t^2(-\nabla_{x_t} U_t(x_t;\eta)) + x_t - D_t(x_t;\theta) \right\|^2$
      Energy Pinning$(\eta) = \nabla_\eta \, \mathbb{E}_x \|(-U_{t=1}(x;\eta)) - \beta_{i+1} \log \pi(x)\|^2$
      $\theta \leftarrow \texttt{FirstOrderOptimizer}(\theta, \text{Score Matching}(\theta), \text{Target Score}(\theta))$
      $\eta \leftarrow \texttt{FirstOrderOptimizer}(\eta, \text{Energy Matching}(\eta), \text{Energy Pinning}(\eta))$
  **end for**
  **return** trained parameters $\theta^*, \eta^*$

---

learn the score function $s_t(x;\theta) \approx \nabla \log p_t(x)$, we follow the standard practice and parameterize the denoising model $D_t(x_t;\theta) = \sigma_t^2 s_t(x_t;\theta) + x_t$, which we learn via the denoising score matching (DSM) objective (Ho et al., 2020), i.e.

$$\text{Denoising Score Matching}(\theta) = \mathbb{E}_{t,x_t,x} \lambda(t)\|x - D_t(x_t;\theta)\|^2, \tag{13}$$

where the expectation is taken w.r.t. samples from the annealed target $x \sim \pi^{\beta+\Delta\beta}(x)$, noised samples $x_t \sim \mathcal{N}(x_t \,|\, \alpha_{1-t}x, \sigma_{1-t}^2)$, and time parameter sampled with $\log(1 - t) \sim \mathcal{N}(P_{mean}, P_{std})$ largely following Karras et al. (2022).

However, the DSM objective is not sufficient for training a good score model close to the target distribution ($\tau = 1 - t = 0$) due to the high variance of the estimator. Indeed, for $\tau = 1 - t = 0$, it has no information about the target distribution. Target Score Matching (De Bortoli et al., 2024) overcomes this issue by explicitly incorporating the score of the target unnormalized density into the objective, which is as follows

$$\text{Target Score Matching}(\theta) = \mathbb{E}_{t,x_t,x} \left[ \left\| \sigma_t^2 \nabla_x \log \pi(x) + x - D_t(x_t;\theta) \right\|^2 \cdot \mathbb{I}(t \geq t_{\text{thresh}}) \right] \tag{14}$$

where the expectation is taken w.r.t. the same random variables as in Eq. (13), but the time variable is restricted to the interval $[t_{\text{thresh}}, 1]$ because the variance of the objective estimator grows with the noise scale (De Bortoli et al., 2024). Notably, for larger noise scales, the Denoising Score Matching objective results in a stable training dynamics; thus, these objectives complement each other and result in a stable training dynamics across the entire time interval.

To train the energy model $U_t(x;\eta)$, which plays the central role in the inference-time annealing (see Section 3.1), we follow Thornton et al. (2025) and distill the learned score model to the parametric energy model via the following regression loss (w.r.t. $\eta$), i.e.

$$\text{EBM Distillation}(\eta) = \mathbb{E}_{t,x_t,x} \lambda(t) \left\| \sigma_t^2(-\nabla_{x_t} U_t(x_t;\eta)) + x_t - D_t(x_t;\theta) \right\|^2, \tag{15}$$

where, the expectation is taken w.r.t. the same random variables as in Eq. (13). Note that, in contrast to the denoising score matching loss in Eq. (13), the "target" in Eq. (15) does not depend on $x$, which means that its variance for the same $x_t$ is zero, stabilizing the training of the energy based model $U_t(x;\eta)$.

Finally, to use all the supervision signal available in the problem, we use the target unnormalized density $\pi(x)^{\beta_{i+1}}$ as the regression target for the end-point energy-based model $U_{t=1}(x;\eta)$, and introduce the following loss

$$\text{Energy Pinning}(\eta) = \mathbb{E}_{\pi^{\beta_{i+1}}(x)} \|(-U_{t=1}(x;\eta)) - \beta_{i+1} \log \pi(x)\|^2. \tag{16}$$

Notably, this loss allows for fixing the gauge present due to the shift invariance of the energy-based model ($\nabla_x(U_t(x;\eta)) = \nabla_x(U_t(x;\eta) + \text{const})$). In practice, we observe that this loss significantly stabilizes the training and improves the final performance. We present the pseudo-code for the training loop in Algorithm 1, where we simultaneously optimize all the introduced loss functions to train a diffusion model at temperature $1/\beta_{i+1}$. In practice, we find that sequential training of models demonstrates the best performance. Furthermore, we initialize the model for the next temperature $1/\beta_{i+1}$ with the parameters of the trained model for the temperature $1/\beta_i$.

# 4 Related work

**Diffusion-based Sampling**. A variety of amortized samplers that use properties of diffusion models have recently been proposed in the literature. Simulation-based approaches that also exploit the fast mode-mixing of diffusion models include Berner et al. (2024); Vargas et al. (2023); Zhang and Chen (2022); Richter et al. (2024); Vargas et al. (2024), which exploit diffusion processes for fast mode mixing. Conversely, simulation-free methods like iDEM (Akhound-Sadegh et al., 2024), SB with Föllmer drift (Huang et al., 2025), and TSM (De Bortoli et al., 2024) offer more scalable approaches to learning the score but suffer from inefficient and high variance score estimates far from the data. Finally, new diffusion bridges have also risen to prominence with underdamped dynamics (Blessing et al., 2024), known modes (Noble et al., 2025), and bridges with SMC (Chen et al., 2025).

**Inference-time Resampling**. The inference-time annealing scheme proposed in Proposition 1 connects several recently proposed methods. Namely, for $\xi_t \equiv 0$, it closely matches the importance sampling of the continuous normalizing flows proposed in (Köhler et al., 2020). Indeed, Eq. (11) becomes a probability flow ODE, and Eq. (12) becomes an integration of the log-density-ratio, where the target density can be defined either as a linear interpolation of log-densities or only in the final point as the target density. Furthermore, Proposition 1 is an application of the Feynman-Kac formula to annealing and non-equilibrium dynamics simultaneously. Indeed, for $\gamma = 1$, this proposition becomes the result proposed in Vaikuntanathan and Jarzynski (2008); Albergo and Vanden-Eijnden (2025); whereas, for $s_t(x) = -\nabla U_t(x; \eta) = \nabla \log p_t(x)$, this proposition becomes the result from (Skreta et al., 2025). In practice, however, these equalities are not satisfied because we use learned models for the vector field $s_t(x) = s_t(x; \theta)$ and the energy-based model $U_t(x; \eta)$. In concurrent work, Rissanen et al. (2025) also propose an annealed sampling scheme using diffusion models. They use a different inference-time annealing procedure which restricts importance re-sampling to the final timestep, whereas in this work, we benefit from *annealed* importance sampling over diffusion time.

**Boltzmann Generators**. Noé et al. (2019) proposed training a probabilistic model and resampling the generated samples according to the target Boltzmann density via importance sampling. Various probabilistic models have been used, e.g. continuous normalizing flows (Chen et al., 2018) and the flow matching objective (Lipman et al., 2023) also allow for efficient training and resampling under the Boltzmann generators framework (Köhler et al., 2020; Klein et al., 2023). Boltzmann Generators can also be combined with Annealed Importance Sampling, which enhances their scalability (Tan et al., 2025a). However, as we demonstrate empirically, the straightforward resampling with a target density of a different temperature (Schopmans and Friederich, 2025) results in high variance of importance weights. Thus, one has to deviate from the Boltzmann Generators framework to perform the inference-time annealing.

# 5 Experiments

We evaluate PITA on molecular conformation sampling tasks including a toy Lennard-Jones system of 13 particles (LJ-13) and Alanine peptide systems of varying sizes (Alanine Dipeptide and Tripeptide) in Cartesian coordinate space. Throughout, we assume access to a short MCMC chain run at high temperature. Note that we do not require these chains to be well mixed, but only require them to cover the modes, a much less stringent requirement (See Appendix D). For metrics, we use sample-based metrics such as 2-Wasserstein distance on Ramachandran coordinates ($\mathbb{T}$-$\mathbb{W}_2$) and energy distribution ($\mathcal{E}$-$\mathbb{W}_1$, $\mathcal{E}$-$\mathbb{W}_2$), to assess mode coverage and precision, respectively. We also compare the KL divergence between the Ramachandran plots of the ground-truth MD samples and the generated samples (RAM-KL), as well as Wasserstein distances on the first two TICA (Time-lagged Independent Component Analysis) coordinates of ground-truth and generated samples. Finally, we report the computational expense of all methods using the total number of energy evaluation calls.

**Baselines**. We compare PITA with three different baselines: Temperature Annealed Boltzmann Generators (TA-BG, Schopmans and Friederich (2025)), normalizing flow model trained on molecular simulation (MD) data collected at the target temperature (MD-NF), diffusion model trained on molecular simulation (MD) data collected at the target temperature (MD-Diff), and importance sampling using continuous normalizing flows (Köhler et al., 2020). For the LJ-13 dataset, we additionally compare the performance of the model with two state-of-the-art diffusion-based sampling algorithms: namely, iDEM (Akhound-Sadegh et al., 2024) and adjoint sampling (Havens et al., 2025); however, as none of the current diffusion-based approaches are able to achieve competitive

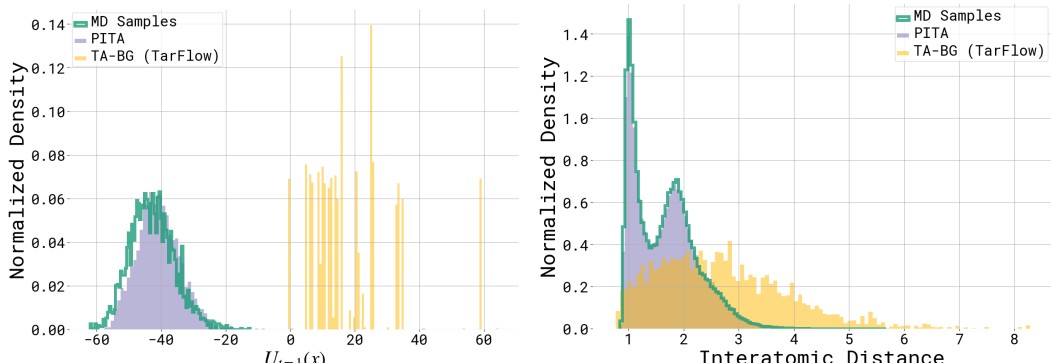

Figure 2: LJ-13 sampling task. We compare the distribution of the interatomic distances and energy of the particles in the MCMC dataset (ground-truth), samples generated using a PITA model, and TA-BG progressively trained from high temperature to sample from the target distribution.

performance on the small protein tasks, we only compare to the TA-BG baseline for those datasets. Additionally, for the ALDP experiment, we compare against two other annealing strategies for the diffusion-only approach: FKC and Score Scaling. FKC refers to the annealing scheme in Skreta et al. (2025), and score scaling is essentially scaling the score with the corresponding temperature factor $\gamma$. Further experimental details of the baselines, as well as additional baselines, are provided in the Appendices E and I.

**Architecture**. In training PITA, we use EGNN (Satorras et al., 2021) as the model backbone for LJ-13, and DiT (Peebles and Xie, 2023) for Alanine Dipeptide and Alanine Tripeptide. Our training follows a sequential temperature schedule, proceeding from high to low temperatures. After training at a given temperature for a fixed number of epochs, we generate samples at the next lower temperature and continue training at that temperature. For LJ-13, we train a single diffusion model conditioning it on $\beta$ and using the data across all previously seen temperatures. For molecular conformation sampling tasks, we adopt a fine-tuning approach, where at each temperature step, the model is trained only on the newly generated samples corresponding to the current temperature without revisiting earlier ones. For the TA-BG baseline, we train TarFlow (Zhai et al., 2025) with adaptations suited to molecular data for all three systems, and for MD-Diff, we use the same DiT architecture we used in PITA. We parameterize the energy network using the parameterization in Thornton et al. (2025) and use the preconditioning ($c_s$, $c_{out}$, $c_{in}$, $c_t$) of Karras et al. (2022) for both energy and score networks. Further training details and hyperparameters are provided in Appendix I.

**Hyperparameters**. Proposition 1 allows for many choices of the vector field $s_t(x)$. In practice, we set it proportional to the score model $s_t(x) \propto s_t(x; \theta)$ and try several scaling coefficients (see Appendix G). Finally, one can easily add the time-dependent schedule $\gamma_t$ by adding the extra term to the weights, i.e. $d \log w_t = $ Eq. (12) $- U_t(x_t; \eta) \partial \gamma_t / \partial t dt$, which we study in Appendix G. We use the noise schedule from Karras et al. (2022), where, for all experiments, we set $\sigma_{\max} = 80$ and $\rho = 7$. For LJ-13 and molecular conformation sampling tasks, we use $\sigma_{\min} = 0.05, 0.01$, respectively.

### 5.1 Main results

**LJ-13.** We first consider a Lennard-Jones (LJ) system of 13 particles to demonstrate the effectiveness of training a sampler at a high temperature ($T_L = 4$), followed by annealing to a lower temperature ($T_S = 1$). As shown in Table 1, we compare the performance of PITA with existing baselines. A visual comparison to TA-BG is provided in Fig. 2. We evaluate each method using the

Table 1: LJ-13 sampling task. The starting temperature is $T_L = 4$, annealed to $T_S = 1$.

| Algorithm | Distance-$\mathcal{W}_2 \downarrow$ | Energy-$\mathcal{W}_2 \downarrow$ | Geometric-$\mathcal{W}_2 \downarrow$ |
|---|---|---|---|
| iDEM | 0.127 | 30.78 ± 24.46 | **1.61 ± 0.01** |
| Adjoint Sampling | - | 2.40 ± 1.25 | 1.67 ± 0.01 |
| TA-BG (TarFlow) | 1.21 ± 0.02 | 61.47 ± 0.12 | 4.16 ± 0.01 |
| **PITA (Ours)** | **0.04 ± 0.00** | **2.26 ± 0.21** | 1.65 ± 0.00 |

2-Wasserstein distance computed over interatomic distance distributions, energy distributions, and 3D geometric coordinates. We omit the Distance-$\mathcal{W}_2$ metric for Adjoint Sampling, as its results could not be reproduced and no code is available at this time; the reported Energy- and Geometric-$\mathcal{W}_2$ values are taken directly from the original paper (Havens et al., 2025). To ensure consistency, we exclude samples with energy above 1000 across all methods; this notably impacts TA-BG, removing

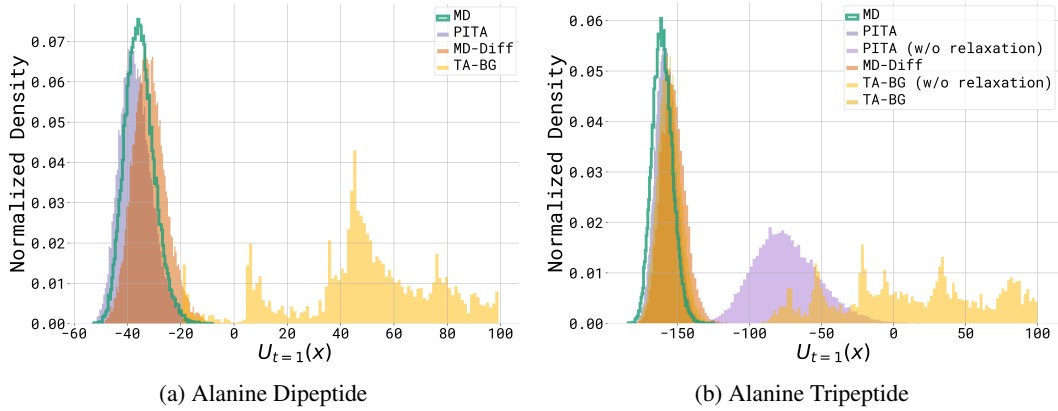

(a) Alanine Dipeptide       (b) Alanine Tripeptide

Figure 3: Molecular conformation sampling tasks. We compare the energy distribution of the ground-truth MD dataset and the samples generated using different models at 300K. We use 30k samples for the plots.

Table 2: Performance of methods for the ALDP sampling task. The starting temperature is $T_L = 1200$ K, annealed to target $T_S = 300$ K. Metrics are calculated over 10k samples and standard deviations over 3 seeds.

| | Rama-KL | Tica-$\mathcal{W}_1 \downarrow$ | Tica-$\mathcal{W}_2 \downarrow$ | Energy-$\mathcal{W}_1 \downarrow$ | Energy-$\mathcal{W}_2 \downarrow$ | $\mathbb{T}$-$\mathcal{W}_2$ | #Energy Evals |
|---|---|---|---|---|---|---|---|
| PITA | $4.773 \pm 0.460$ | $\mathbf{0.112 \pm 0.006}$ | $\mathbf{0.379 \pm 0.028}$ | $1.530 \pm 0.068$ | $1.615 \pm 0.053$ | $\mathbf{0.270 \pm 0.023}$ | $5 \times 10^7$ |
| MD-Diff | $\mathbf{1.308 \pm 0.072}$ | $0.113 \pm 0.001$ | $0.579 \pm 0.004$ | $3.627 \pm 0.023$ | $3.704 \pm 0.026$ | $0.310 \pm 0.001$ | $5 \times 10^7$ |
| MD-NF | $13.533 \pm 0.024$ | $0.138 \pm 0.003$ | $0.586 \pm 0.003$ | $\mathbf{0.551 \pm 0.062}$ | $\mathbf{1.198 \pm 0.069}$ | $0.403 \pm 0.045$ | $5 \times 10^7$ |
| TA-BG | $14.993 \pm 0.002$ | $0.219 \pm 0.013$ | $0.685 \pm 0.034$ | $83.457 \pm 0.070$ | $86.176 \pm 0.104$ | $0.979 \pm 0.012$ | $5 \times 10^7$ |
| FKC | $14.392 \pm 0.909$ | $0.217 \pm 0.000$ | $0.649 \pm 0.001$ | $11.281 \pm 0.025$ | $11.466 \pm 0.027$ | $2.120 \pm 0.024$ | $5 \times 10^7$ |
| Score Scaling | $4.588 \pm 0.467$ | $0.183 \pm 0.002$ | $0.608 \pm 0.008$ | $10.282 \pm 0.020$ | $10.460 \pm 0.019$ | $0.550 \pm 0.036$ | $5 \times 10^7$ |

Table 3: Performance of methods for the AL3 sampling task. The starting temperature is $T_L = 1200$ K, annealed to target $T_S = 300$K. Metrics are calculated over 10k samples and standard deviations over 3 seeds.

| | Rama-KL | Tica-$\mathcal{W}_1 \downarrow$ | Tica-$\mathcal{W}_2 \downarrow$ | Energy-$\mathcal{W}_1 \downarrow$ | Energy-$\mathcal{W}_2 \downarrow$ | $\mathbb{T}$-$\mathcal{W}_2$ | #Energy Evals |
|---|---|---|---|---|---|---|---|
| PITA | $\mathbf{1.209 \pm 0.144}$ | $0.272 \pm 0.017$ | $0.952 \pm 0.055$ | $\mathbf{2.567 \pm 0.108}$ | $\mathbf{2.592 \pm 0.107}$ | $0.521 \pm 0.006$ | $8 \times 10^7$ |
| PITA (w/o relaxation) | $8.535 \pm 0.254$ | $0.405 \pm 0.014$ | $0.999 \pm 0.043$ | $86.270 \pm 0.294$ | $87.695 \pm 0.294$ | $0.651 \pm 0.013$ | $5 \times 10^7$ |
| MD-Diff | $9.662 \pm 0.085$ | $\mathbf{0.059 \pm 0.006}$ | $\mathbf{0.426 \pm 0.010}$ | $7.416 \pm 0.130$ | $7.599 \pm 0.137$ | $0.424 \pm 0.011$ | $8 \times 10^7$ |
| TA-BG | $2.078 \pm 2.088$ | $0.082 \pm 0.001$ | $0.454 \pm 0.001$ | $4.782 \pm 0.076$ | $4.863 \pm 0.082$ | $\mathbf{0.347 \pm 0.014}$ | $8 \times 10^7$ |
| TA-BG (w/o relaxation) | $14.988 \pm 0.009$ | $0.321 \pm 0.001$ | $0.648 \pm 0.000$ | $173.042 \pm 0.717$ | $178.558 \pm 0.732$ | $1.310 \pm 0.004$ | $5 \times 10^7$ |

approximately 60% of its samples. Even under this filtering, PITA consistently outperforms TA-BG and other baselines trained directly at the target temperature.

**Alanine Dipeptide**. We apply PITA to the task of sampling Alanine Dipeptide at a target temperature of $T_S = 300$ K, given initial samples at a higher temperature of $T_L = 1200$ K. We use annealing steps of 1200 K, 755.95 K, 555.52 K, 300 K. These temperatures correspond to a subset of the temperatures from Schopmans and Friederich (2025), as PITA does not require as many annealing steps to achieve competitive performance. We also analyze the performance of the model, taking larger annealing steps in Appendix G. As shown in Table 2, PITA consistently outperforms both the diffusion-based baseline and TA-BG across all evaluation metrics, achieving a particularly large margin in energy-related metrics. We further present TICA plots of the generated samples at the target temperature in Fig. 4. PITA successfully recovers the essential slow collective dynamical modes of the system, which baseline methods fail to capture. Additionally, we find that while TA-BG performs reasonably well at earlier stages of training at higher temperatures, its performance deteriorates as temperature decreases. Such a decline is likely due to the increasing difficulty in generating high-quality proposals as the temperature decreases, which is crucial in the importance sampling used for subsequent training stages, as well as the difficulty in importance sampling between large temperature gaps (Appendix D.2). Additional details on training dynamics across all temperatures for PITA and TA-BG can be found in Appendix H.

**Alanine Tripeptide.** We further evaluate the performance of PITA on a larger molecular system, Alanine Tripeptide (AL3). We employ a temperature annealing schedule with intermediate steps at 1200 K, 755.95 K, 555.52 K, 408.24 K, 300 K. As shown in Fig. 5, PITA again successfully recovers the essential dynamical modes of the system, indicating its capability of generating samples that align with the dominant kinetic features of the underlying dynamics. In practice, we also observe that performing a short additional MD refinement at the target temperature (300 K) after generation

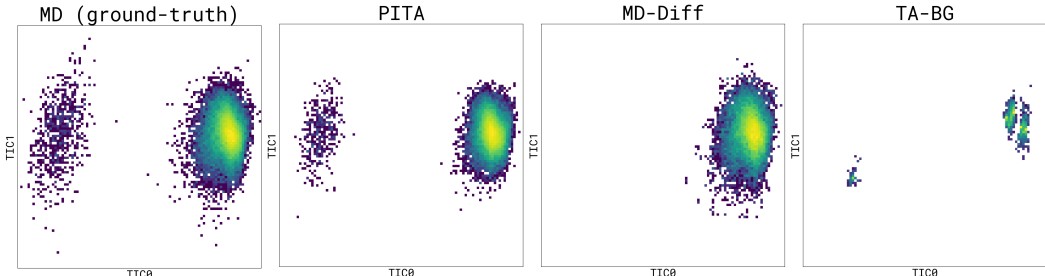

Figure 4: TICA plots for Alanine Dipeptide (ALDP) at 300K obtained from different methods using 30k samples. Each panel shows the free energy landscape along the top two TICA components which capture the dominant slow transitions in the system.

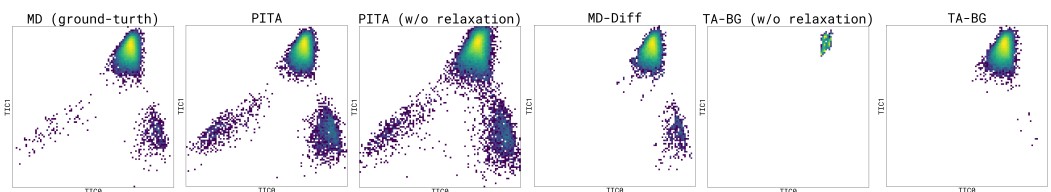

Figure 5: TICA plots for Alanine Tripeptide (AL3) at 300K obtained from different methods using 30k samples.

further improves the physical plausibility and smoothness of the trajectories. This leads to more accurate estimates of the free energy landscape, for both PITA and TA-. In Table 3, we provide quantitative analysis of the performance of the models. Notably, despite resulting in a better mode coverage, PITA performs worse than baselines according to Tica$-\mathcal{W}_1$ and Tica$-\mathcal{W}_2$, which suggest that it does not fully recover the correct relative weights of the modes.

## 6  Conclusion

In this paper, we propose PITA a new framework to train diffusion-based samplers by introducing two mechanisms of interpolating a target Boltzmann density by changing the temperature and defining a diffusion noising process. We demonstrated that PITA allows the progressive training of a sequence of diffusion models that go from high temperature, where ground truth data is simple to collect, to the lower temperature target temperature. Using PITA we demonstrated equilibrium sampling of $N$-body particle systems, and, for the first time, equilibrium sampling of alanine dipeptide and tripeptide in Cartesian coordinates. Importantly, we demonstrate PITA requires drastically fewer energy evaluations than existing diffusion samplers.

We believe that PITA represents a step forward in the scalability of diffusion-based samplers and opens up ripe avenues for future work including improved training strategies and regimes for energy-based models that are in service of the PITA framework. Natural directions for future work include automatically determining the optimal temperature jump when instantiating our Feynman-Kac PDE to generate asymptotically unbiased samples at lower temperatures or transferable sampling (Klein and Noé, 2024; Tan et al., 2025b).

**Limitations**. To obtain a consistent estimator or an importance sampling scheme, one has to define a density model of the generated samples. For this, PITA relies on training an additional energy-based model, which is a notoriously challenging task. Furthermore, simultaneous training and inference of both the score model and the energy-based model introduces additional computational and memory requirements.

## Acknowledgments

This project was partially sponsored by Google through the Google & Mila projects program. The authors acknowledge funding from UNIQUE, CIFAR, NSERC, Intel, and Samsung. The research was enabled in part by computational resources provided by the Digital Research Alliance of Canada (https://alliancecan.ca), Mila (https://mila.quebec), and NVIDIA. KN was supported by IVADO and Institut Courtois. AJB is partially supported by an NSERC Post-doc fellowship. This research is partially supported by the EPSRC Turing AI World-Leading Research Fellowship No. EP/X040062/1 and EPSRC AI Hub No. EP/Y028872/1.

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

# A Proofs

## A.1 Inference-Time Annealing

**Proposition 1.** [Inference-time Annealing] *Annealed density of the energy-based model $q_t(x) \propto \exp(-\gamma U_t(x; \eta))$ matches the marginal densities of the following SDE*

$$dx_t = \left(-a_t x_t + \frac{\zeta_t^2}{2}(s_t(x_t) - \gamma \xi_t \nabla U_t(x_t; \eta))\right) dt + \zeta_t \sqrt{\xi_t} dW_t , \quad x_0 \sim q_{t=0}(x) \quad (11)$$

$$d \log w_t = \left[\frac{\zeta_t^2}{2} \langle \nabla, s_t(x_t) \rangle - \gamma \left\langle \nabla U_t(x_t; \eta), -a_t x_t + \frac{\zeta_t^2}{2} s_t(x_t) \right\rangle - \gamma \frac{\partial U_t(x_t; \eta)}{\partial t}\right] dt , \quad (12)$$

*where $s_t(x)$ is any vector field, $a_t, \zeta_t, \xi_t$ are analogous to parameters from Eq. (4), and the sample weights $w_t$ correspond to the SNIS estimator of the Feynman-Kac formula in Eq. (9).*

*Proof.* For the Energy-Based Model $U_t(x; \eta)$, we denote the corresponding Boltzmann density as

$$q_t(x) \propto \exp(-\gamma U_t(x; \eta)), \quad (17)$$

where $\gamma$ is the target inverse temperature. Taking the time-derivative of $q_t(x)$ we get the following equation

$$\frac{\partial q_t(x)}{\partial t} = q_t(x) \left(-\gamma \frac{\partial U_t(x; \eta)}{\partial t} + \mathbb{E}_{q_t(x)} \gamma \frac{\partial U_t(x; \eta)}{\partial t}\right), \quad (18)$$

which can be simulated by reweighting the samples from $q_{t=0}(x) \approx \mathcal{N}(0, 1)$ according to the following weights

$$w(x_1) = \frac{\exp(-\gamma \int_0^1 dt \; \partial U_t(x_t; \eta)/\partial t)}{\mathbb{E}_{q_0(x_0)} \exp(-\gamma \int_0^1 dt \; \partial U_t(x_t; \eta)/\partial t)} . \quad (19)$$

Although this reasoning is theoretically justified, in practice, the variance of this importance-weighted estimate (or resampled distribution) is prohibitively large. That is why one has to introduce additional terms that move samples around tracing the original diffusion process. Namely, we consider the following PDE

$$\frac{\partial q_t(x)}{\partial t} = \pm \left\langle \nabla, q_t(x)(-a_t x + \frac{\zeta_t^2}{2} s_t(x)) \right\rangle + q_t(x) \left(-\gamma \frac{\partial U_t(x; \eta)}{\partial t} + \mathbb{E}_{q_t(x)} \gamma \frac{\partial U_t(x; \eta)}{\partial t}\right)$$

$$= -\left\langle \nabla, q_t(x)(-a_t x + \frac{\zeta_t^2}{2} s_t(x)) \right\rangle + q_t(x) \big(g_t(x) - \mathbb{E}_{q_t(x)} g_t(x)\big), \quad (20)$$

$$g_t(x) = \gamma \langle \nabla U_t(x; \eta), a_t x \rangle - \gamma \frac{\zeta_t^2}{2} \langle \nabla U_t(x; \eta), s_t(x) \rangle - \langle \nabla, a_t x \rangle + \quad (21)$$

$$+ \frac{\zeta_t^2}{2} \langle \nabla, s_t(x) \rangle - \gamma \frac{\partial U_t(x; \eta)}{\partial t} , \quad (22)$$

where the term $\langle \nabla, a_t x \rangle$ does not depend on $x$ and cancels out when in the reweighting term. Furthermore, we can introduce the noise term by adding and subtracting the score $\xi_t(\zeta_t^2/2) \nabla \log q_t(x)$, i.e.

$$\frac{\partial q_t(x)}{\partial t} = -\left\langle \nabla, q_t(x)(-a_t x + \frac{\zeta_t^2}{2} s_t(x) - \gamma \xi_t \frac{\zeta_t^2}{2} \nabla U_t(x; \eta)) \right\rangle + \xi_t \frac{\zeta_t^2}{2} \Delta q_t(x) + \quad (23)$$

$$+ q_t(x) \big(g_t(x) - \mathbb{E}_{q_t(x)} g_t(x)\big), \quad (24)$$

$$g_t(x) = -\gamma \left\langle \nabla U_t(x; \eta), -a_t x + \frac{\zeta_t^2}{2} s_t(x) \right\rangle + \frac{\zeta_t^2}{2} \langle \nabla, s_t(x) \rangle - \gamma \frac{\partial U_t(x; \eta)}{\partial t} , \quad (25)$$

which can be simulated as

$$dx_t = \left(-a_t x_t + \frac{\zeta_t^2}{2}(s_t(x_t) - \gamma \xi_t \nabla U_t(x_t; \eta))\right) dt + \zeta_t \sqrt{\xi_t} dW_t , \quad (26)$$

$$d \log w_t = \left[-\gamma \left\langle \nabla U_t(x_t; \eta), -a_t x_t + \frac{\zeta_t^2}{2} s_t(x_t) \right\rangle + \frac{g_t^2}{2} \langle \nabla, s_t(x_t) \rangle - \gamma \frac{\partial U_t(x_t; \eta)}{\partial t}\right] dt . \quad (27)$$

$\square$

**Proposition 2.** [Convergence to Diffusion] *For $\gamma = 1$ and perfect models $s_t(x) = -\nabla U_t(x; \eta) = \nabla \log p_t(x)$, the variance of the weights in Proposition 1 becomes zero.*

*Proof.* Indeed, for $\gamma = 1$, Eq. (25) becomes

$$g_t(x) = -\left\langle \nabla U_t(x; \eta), -a_t x + \frac{\zeta_t^2}{2} s_t(x) \right\rangle + \frac{\zeta_t^2}{2} \langle \nabla, s_t(x) \rangle - \frac{\partial U_t(x; \eta)}{\partial t} \tag{28}$$

$$= \left\langle \nabla U_t(x; \eta), a_t x - \frac{\zeta_t^2}{2} s_t(x) \right\rangle - \left\langle \nabla, a_t x_t - \frac{\zeta_t^2}{2} s_t(x) \right\rangle + da_t - \frac{\partial U_t(x; \eta)}{\partial t},$$

where $d$ is the dimensionality of the state-space. For $s_t(x) = -\nabla U_t(x; \eta) = \nabla \log p_t(x)$, we have

$$g_t(x) = -\frac{1}{p_t(x)} \left\langle \nabla, p_t(x) \left( a_t x_t - \frac{\zeta_t^2}{2} \nabla \log p_t(x) \right) \right\rangle + \frac{\partial \log p_t(x)}{\partial t} + \frac{\partial \log Z_t}{\partial t} + da_t \tag{29}$$

$$= \frac{1}{p_t(x)} \underbrace{\left[ -\left\langle \nabla, p_t(x) \left( a_t x - \frac{\zeta_t^2}{2} \nabla \log p_t(x) \right) \right\rangle + \frac{\partial p_t(x)}{\partial t} \right]}_{=0 \text{ due to Eq. (3)}} + \frac{\partial \log Z_t}{\partial t} + da_t, \tag{30}$$

where $Z_t = \int dx\, p_t(x)$. The term in the brackets equals zero due to Eq. (3) since the ground true marginals $p_t(x)$ are defined as the marginals of the diffusion process. Hence, we have

$$g_t(x) = \frac{\partial \log Z_t}{\partial t} + da_t = \text{ constant of } x, \tag{31}$$

which becomes zero after the normalization $g_t(x) - \mathbb{E}_{q_t(x)} g_t(x)$, which concludes the proof. □

### A.2 Inference-Time Geometric Averaging

For the diffusion process with marginals $p_t(x)$ and the target distribution $p_{t=1}(x) \propto \mathcal{N}(x \,|\, 0, \mathbb{1})^{(1-\beta_i)} \pi(x)^{\beta_i}$, we assume having the energy model $U_t(x; \eta)$ and the score model $s_t(x; \theta)$. Then, for the following density

$$q_t(x) \propto \exp\left( -\frac{\beta_{i+1}}{\beta_i} U_t(x; \eta) - \frac{\beta_{i+1} - \beta_i}{\beta_i} \log \mathcal{N}(x \,|\, 0, (\alpha_{1-t}^2 + \sigma_{1-t}^2)\mathbb{1}) \right), \tag{32}$$

we have $q_{t=1}(x) \approx \mathcal{N}(x \,|\, 0, \mathbb{1})^{(1-\beta_{i+1})} \pi(x)^{\beta_{i+1}}$. To sample from this density, we derive another SDE that performs inference-time geometric averaging. Analogously to Proposition 1, for $\gamma = 1$ and perfectly trained models, the weights become constant, and this SDE yields the reverse-time diffusion SDE.

**Proposition 3.** [Inference-time Geometric Averaging] *For the geometric averaging of the energy-based model $q_t(x) \propto \exp\big((1-\gamma)(-U_{t,\beta}(x, \eta)) + \gamma \log \mathcal{N}(x \,|\, 0, \sigma_t^2)\big)$, the weighted samples from $q_t(x)$ can be collected by running the following SDE*

$$dx_t = -a_t x_t + (1-\gamma)\frac{\zeta_t^2}{2}(s_t(x_t) - \xi_t \nabla U_t(x_t; \eta)) - \gamma\frac{1}{\sigma_t^2}\left(1 + \xi_t \frac{\zeta_t^2}{2}\right)x_t + \zeta_t \sqrt{\xi_t} dW_t,$$

$$d \log w_t = \left\langle -(1-\gamma)\nabla U_t(x_t; \eta) - \gamma\frac{1}{\sigma_t^2}x_t, -a_t x_t + (1-\gamma)\frac{\zeta_t^2}{2}s_t(x_t) - \gamma\frac{1}{\sigma_t^2}x_t \right\rangle + \tag{33}$$

$$+ (1-\gamma)\frac{\zeta_t^2}{2}\langle \nabla, s_t(x_t) \rangle + (1-\gamma)\frac{\partial U_t(x_t; \eta)}{\partial t} + \gamma\frac{1}{\sigma_t^3}\|x_t\|^2\frac{\partial \sigma_t}{\partial t}. \tag{34}$$

*where $s_t(x)$ is any vector field. Finally, unweighted samples from $q_t(x)$ can be sampled using SNIS from Eq. (9).*

*Proof.* For the Energy-Based Model $U_t(x; \eta)$, we denote the corresponding geometric averaged density as

$$q_t(x) \propto \exp\big((1-\gamma)(-U_t(x; \eta)) + \gamma \log \mathcal{N}(x \,|\, 0, \sigma_t^2)\big), \tag{35}$$

where $\gamma$ is the target inverse temperature. Taking the time-derivative of $q_t(x)$ we get the following equation

$$\frac{\partial q_t(x)}{\partial t} = q_t(x)\big(g_t(x) - \mathbb{E}_{q_t(x)}g_t(x)\big), \tag{36}$$

$$g_t(x) = (1-\gamma)\frac{\partial U_t(x;\eta)}{\partial t} + \gamma\frac{1}{\sigma_t^3}\|x\|^2\frac{\partial\sigma_t}{\partial t}. \tag{37}$$

Assuming that the change of the density is close to the trained diffusion process, we introduce the drift-term corresponding to the score of the marginals

$$\frac{\partial q_t(x)}{\partial t} = \underbrace{\pm\left\langle \nabla, q_t(x)(-a_t x + (1-\gamma)\frac{\zeta_t^2}{2}s_t(x) - \gamma\frac{1}{\sigma_t^2}x)\right\rangle}_{\text{fictitious term}} + q_t(x)\big(g_t(x) - \mathbb{E}_{q_t(x)}g_t(x)\big),$$

$$g_t(x) = (1-\gamma)\frac{\partial U_t(x;\eta)}{\partial t} + \gamma\frac{1}{\sigma_t^3}\|x\|^2\frac{\partial\sigma_t}{\partial t}. \tag{38}$$

Moving the positive term to the weights and interpreting the negative term as the continuity equation, we get

$$\frac{\partial q_t(x)}{\partial t} = -\left\langle \nabla, q_t(x)(-a_t x + (1-\gamma)\frac{\zeta_t^2}{2}s_t(x) - \gamma\frac{1}{\sigma_t^2}x)\right\rangle + q_t(x)\big(g_t(x) - \mathbb{E}_{q_t(x)}g_t(x)\big),$$

$$g_t(x) = \left\langle -(1-\gamma)\nabla U_t(x;\eta) - \gamma\frac{1}{\sigma_t^2}x, -a_t x + (1-\gamma)\frac{\zeta_t^2}{2}s_t(x) - \gamma\frac{1}{\sigma_t^2}x\right\rangle + \tag{39}$$

$$+ (1-\gamma)\frac{\zeta_t^2}{2}\langle\nabla, s_t(x)\rangle + (1-\gamma)\frac{\partial U_t(x;\eta)}{\partial t} + \gamma\frac{1}{\sigma_t^3}\|x\|^2\frac{\partial\sigma_t}{\partial t}. \tag{40}$$

Finally, we introduce the noise term by adding the drift

$$\xi_t\frac{\zeta_t^2}{2}\nabla\log q_t(x) = \xi_t\frac{\zeta_t^2}{2}\left(-(1-\gamma)\nabla U_t(x;\eta) - \gamma\frac{1}{\sigma_t^2}x\right). \tag{41}$$

Thus, we get

$$\frac{\partial q_t(x)}{\partial t} = -\left\langle \nabla, q_t(x)\left(-a_t x + (1-\gamma)\frac{\zeta_t^2}{2}(s_t(x) - \xi_t\nabla U_t(x;\eta)) - \gamma\frac{1}{\sigma_t^2}\left(1 + \xi_t\frac{\zeta_t^2}{2}\right)x\right)\right\rangle +$$

$$+ \xi_t\frac{\zeta_t^2}{2}\Delta q_t(x) + q_t(x)\big(g_t(x) - \mathbb{E}_{q_t(x)}g_t(x)\big), \tag{42}$$

$$g_t(x) = \left\langle -(1-\gamma)\nabla U_t(x;\eta) - \gamma\frac{1}{\sigma_t^2}x, -a_t x + (1-\gamma)\frac{\zeta_t^2}{2}s_t(x) - \gamma\frac{1}{\sigma_t^2}x\right\rangle + \tag{43}$$

$$+ (1-\gamma)\frac{\zeta_t^2}{2}\langle\nabla, s_t(x)\rangle + (1-\gamma)\frac{\partial U_t(x;\eta)}{\partial t} + \gamma\frac{1}{\sigma_t^3}\|x\|^2\frac{\partial\sigma_t}{\partial t}. \tag{44}$$

The corresponding SDE is

$$dx_t = -a_t x_t + (1-\gamma)\frac{\zeta_t^2}{2}(s_t(x_t) - \xi_t\nabla U_t(x_t;\eta)) - \gamma\frac{1}{\sigma_t^2}\left(1 + \xi_t\frac{\zeta_t^2}{2}\right)x_t + \zeta_t\sqrt{\xi_t}dW_t,$$

$$d\log w_t = \left\langle -(1-\gamma)\nabla U_t(x_t;\eta) - \gamma\frac{1}{\sigma_t^2}x_t, -a_t x_t + (1-\gamma)\frac{\zeta_t^2}{2}s_t(x_t) - \gamma\frac{1}{\sigma_t^2}x_t\right\rangle + \tag{45}$$

$$+ (1-\gamma)\frac{\zeta_t^2}{2}\langle\nabla, s_t(x_t)\rangle + (1-\gamma)\frac{\partial U_t(x_t;\eta)}{\partial t} + \gamma\frac{1}{\sigma_t^3}\|x_t\|^2\frac{\partial\sigma_t}{\partial t}. \tag{46}$$

$$\square$$

# B   Bridging the Gap at the End-Point

Integrating the dynamics from Propositions 1 and 3 we generate a set of weighted samples $\{(x_{t=1}^k, w_{t=1}^k)\}_{k=1}^K$ that converge to the samples from $q_{t=1}(x)$ when $K \to \infty$. In Section 3.1

we assume that this density is defined as the Boltzmann distribution of the corresponding energy model, i.e. $q_{t=1}(x) \propto \exp(-\beta_{i+1}/\beta_i \cdot U_{t=1}(x; \eta))$, which approximates $\pi^{\beta_{i+1}}$, but does not necessarily match it exactly. Here we describe two possible ways to bridge the gap between the density model and the target density.

The first way to sample from $\pi(x)^{\beta_{i+1}}$ is via Self-Normalized Importance Sampling (SNIS). The integrated weights $w_{t=1}^k = e^{\int_0^1 dt\, g_t(x_t)}$ represent the density ratio between the distribution of the samples $x_{t=1}^k$ and the density of the integrated PDE (see discussion in Section 2.3). Correspondingly, to sample from $\pi(x)^{\beta_{i+1}}$, we have to take into account the density ratio $\pi(x)^{\beta_{i+1}}/q_{t=1}(x)$ and obtain a new estimator, i.e.

$$\mathbb{E}_{\pi(x)^{\beta_{i+1}}} \varphi(x) \propto \mathbb{E}_{q_1(x)} \frac{\pi(x)^{\beta_{i+1}}}{q_1(x)} \varphi(x) \propto \mathbb{E}\left[ e^{\int_0^1 dt\, g_t(x_t)} \frac{\pi(x_1)^{\beta_{i+1}}}{q_1(x_1)} \varphi(x_1) \right], \tag{47}$$

$$\mathbb{E}_{\pi(x)^{\beta_{i+1}}} \varphi(x) \approx \sum_{k=1}^{K} \frac{\tilde{w}_1^k}{\sum_{j=1}^n \tilde{w}_1^j} \varphi(x_1^k), \; \tilde{w}_1^k := e^{\int_0^1 dt\, g_t(x_t)} \pi(x_1^k)^{\beta_{i+1}}/q_1(x_1^k), \tag{48}$$

where the new weights $\tilde{w}_1^k$ are obtained from the old ones $w_1^k$ by multiplication with the corresponding density ratio. Note, that the following empirical distribution approximates the target density $\pi(x)^{\beta_{i+1}}$

$$\tilde{\pi}(x)^{\beta_{i+1}} = \sum_{k=1}^{K} \frac{\tilde{w}_1^k}{\sum_{j=1}^n \tilde{w}_1^j} \delta(x - x_1^k). \tag{49}$$

The alternative to importance sampling with the density model proposal is the gradual interpolation between the density model and the target during the integration. In particular, one can satisfy the boundary conditions by defining a smooth interpolant between the boundary densities $p_{t=0} = \mathcal{N}(0, I)$, $p_{t=1} = \pi^{\beta_{i+1}}$ and the annealed density model as follows

$$q_t(x) \propto \exp\Bigg[ -\gamma\left(1 - \frac{t}{t_1}\right)_+^\kappa \log \mathcal{N}(0,1) - \left(1 - \frac{1-t}{1-t_2}\right)_+^\kappa \beta_{i+1} \log \pi(x) - \tag{50}$$

$$-\left(1 - \left(1 - \frac{t}{t_1}\right)_+^\kappa - \left(1 - \frac{1-t}{1-t_2}\right)_+^\kappa\right) \frac{\beta_{i+1}}{\beta_i} U_t(x; \eta) \Bigg], \tag{51}$$

where $(x)_+ = \max\{0, x\}$, $0 < t_1 < t_2 < 1$ are the hyperparameters that define switch times between models, and $\kappa$ is the smoothness parameter. Thus, we guarantee that $q_{t=1}(x) \propto \pi(x)^{\beta_{i+1}}$. However, in practice, we found that this interpolation technique results in a high variance of importance weights.

## C  Network Parameterization and Preconditioning

We condition our score network $s$ and our energy network $U_t$ based on findings in EDM (Karras et al., 2022), use an energy parameterization based on Neklyudov et al. (2023) and Thornton et al. (2025), and include a new pre-conditioning on $\beta$. All of our networks are based on a backbone $F_\theta(x_t, t, \beta)$ : $(\mathbb{R}^d \times [0, \infty) \times [1, \infty)) \to \mathbb{R}^d$ is a flexible network architecture based on a diffusion transformer (DiT) backbone (Peebles and Xie, 2023). Specifically, we parameterize our denoiser network $D_\theta$ as:

$$D_\theta(x_t, t, \beta) := (1 + \beta(c_{skip}(t) - 1)x_t + \beta c_{out}(t) F_\theta(c_{in}(t)x_t, c_{noise}(t)) \tag{52}$$

which allows us to define our score network $s_\theta$ as

$$s_\theta(x_t, t, \beta) := \frac{D_\theta(x_t, t, \beta) - x_t}{\sigma_t^2} \tag{53}$$

We pre-condition the energy as

$$U_\eta(x_t, t, \beta) := \beta\left( \frac{1 - a_t c_{skip}(t)}{2\sigma_t^2} \|x_t\|^2 - \frac{\xi_t c_{out}(t)}{c_{in}(t)\sigma_t^2} (x_t \cdot F_\eta(c_{in}(t)x_t, c_{noise}(t))) \right) \tag{54}$$

# D  Molecular Dynamics Analysis

## D.1  MD mode mixing across temperatures

In Fig. 6 and Fig. 7, we analyze the mixing behaviour of MD simulations for ALDP and AL3 across various annealing temperatures. Specifically, we examine simulations consisting of 50 million steps—matching the quantity of MD data used for training PITA at 1200K. As the temperature decreases, the sampling quality deteriorates: the chains exhibit poorer mixing and fail to explore significant regions of the configuration space, missing major modes of the distribution. This is shown both in Ramachandran and TICA plots, as well as the trace plots of the internal angle $\phi$ and the second TICA axis. More specifically, for ALDP, we see that the chain switches out of the main mode 5.8%, 3.0%, 1.0% and 0% of the time at temperatures 1200K, 755.95K, 555.52K and 300K, respectively. For AL3, this happens at a rate of 12.7%, 9.2%, 5.8% and 0%.

This motivates training at a higher temperature then annealing to a lower temperature as is done in PITA. As we are able to take advantage of relatively quick mode mixing at higher temperatures and the ability of inference time annealing to recover samples from a lower temperature.

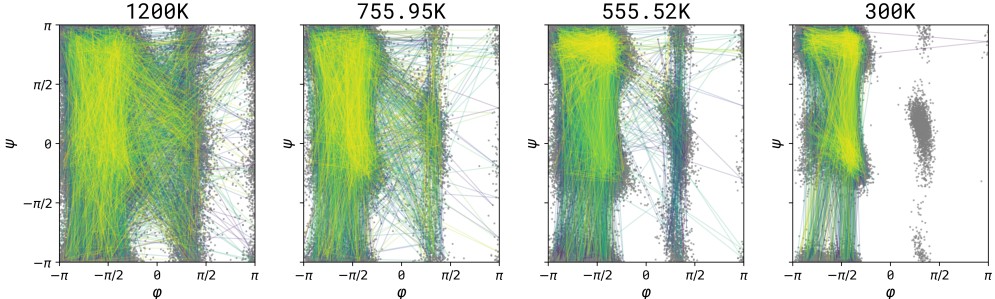

(a) Visualization of the Ramachandran plots of the MD chain over time, the lines are colored from purple to yellow over 50 million MD steps. The gray plot shows the Ramachandran plot of a chain with 1 billion MD steps.

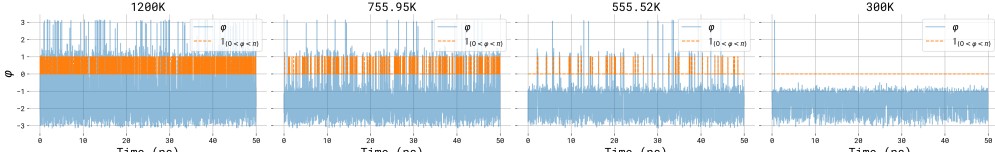

(b) Trace plot for the internal angle $\phi$. The blue lines show the value of $\phi$ across the 50-million-step MD chain. The orange line indicates when the chain switches out of the main modes.

Figure 6: Analysis of the mixing of the MD chains for ALDP, for a 50 million-step MD simulation, across the annealing temperatures.

## D.2  Temperature Annealing ESS

To show the difficulty of directly performing importance sampling directly across large temperature jumps in coordinate space using methods similar to Schopmans and Friederich (2025); Rissanen et al. (2025) we calculate the ESS assuming these models could learn the source temperature perfectly. To perform this experiment we take 33k samples from the test chain and calculate the (normalized) effective sample size of these points when importance sampled to a lower target energy. The results of this experiment are presented in Fig. 8 for ALDP and Fig. 9 for AL3. Here we report the Log normalized effective sample size for each temperature jump considered in this paper. We compute the log normalized ESS using Kish's formula normalized by the number of samples as:

$$\log \mathrm{ESS}\big(\{w_i\}_{i=1}^N\big) = \log \frac{\left(\sum_{i=1}^N w_i\right)^2}{N \sum_{i=1}^N w_i^2}. \tag{55}$$

Where $w_i$ is the importance weight of the $i$th sample. We note that a log ESS of $-5$ would mean that direct importance sampling would require $100\,000$ times as many points at the higher temperature as

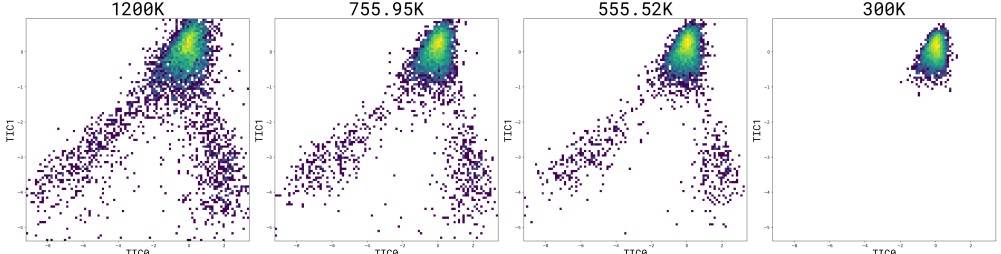

(a) AL3 TICA plots of ground-truth MD samples across different temperatures, using MD chains of 50 million steps. For all temperatures, the TICA axes are matched to those of 300K.

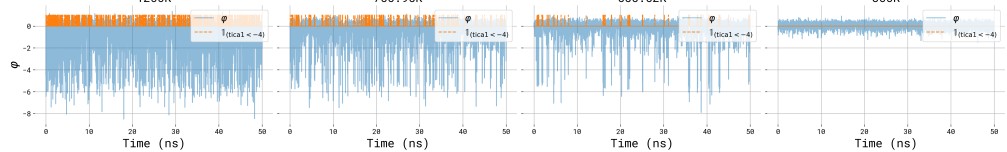

(b) Trace plot for the second TICA axis of ground-truth MD samples. The blue lines show the value of the second TICA axis across the 50-million-step MD chain. The orange line indicates when the chain switches out of the main mode.

Figure 7: Analysis of the mixing of the MD chains for AL3, for a 50 million-step MD simulation, across the annealing temperatures.

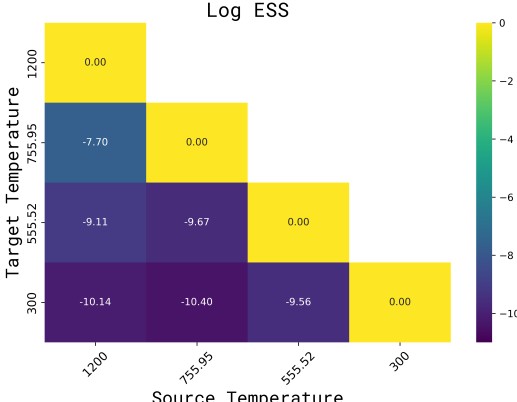

Figure 8: Log effective sample size (ESS) values for importance sampling from different source and target temperatures for ALDP. The ESS values are computed on 33000 samples from the MD chain. The low values indicate that purely relying on importance sampling is not sufficient to sample from lower temperature targets.

are needed at the lower temperature. This makes importance sampling computationally infeasible for these temperature jumps, even with perfectly learned models.

This motivates the guided approach taken in PITA, where we are able to guide the samples preferentially towards the target temperature, avoiding the problem of low ESS between these temperature jumps.

# E  Additional Baselines

For the ALDP experiment, we compare against two additional baselines of annealed MCMC with sequential Monte Carlo (SMC) and parallel tempering (PT) strategies. For SMC, we take 10 annealing temperatures between 1200 K and 300 K using a geometric schedule, taking 166 steps per temperature with 30k particles. For PT, we use the same 4 annealing temperatures as PITA, running 25 MD steps between particle exchanges for a total of 50k iterations. As it can be seen in Table 4, PITA performs better overall in capturing the distribution given the same budget of energy evaluation. In Table 5, we

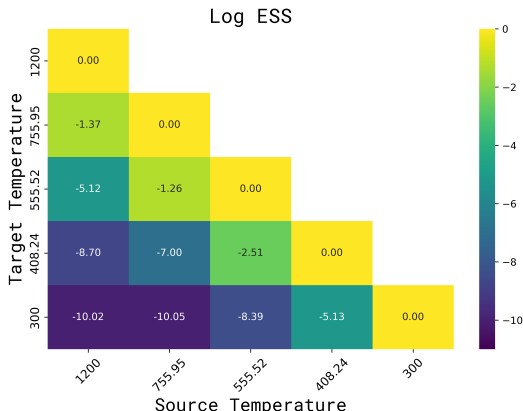

Figure 9: Log effective sample size (ESS) values for importance sampling from different source and target temperatures for AL3. The ESS values are computed on 33000 samples from the MD chain. The low values indicate that purely relying on importance sampling is not sufficient to sample from lower temperature targets.

Table 4: Performance of the SMC and PT baselines on ALDP

|  | Rama-KL | Tica-$\mathcal{W}_1 \downarrow$ | Tica-$\mathcal{W}_2 \downarrow$ | Energy-$\mathcal{W}_1 \downarrow$ | Energy-$\mathcal{W}_2 \downarrow$ | $\mathbb{T}$-$\mathcal{W}_2$ | #Energy Evals |
|---|---|---|---|---|---|---|---|
| PITA | $4.773 \pm 0.460$ | $0.112 \pm 0.006$ | $0.379 \pm 0.028$ | $1.530 \pm 0.068$ | $1.615 \pm 0.053$ | $0.270 \pm 0.023$ | $5 \times 10^7$ |
| PT | $7.306 \pm 1.077$ | $0.625 \pm 0.010$ | $0.895 \pm 0.016$ | $4.652 \pm 0.015$ | $4.689 \pm 0.014$ | $0.911 \pm 0.004$ | $5 \times 10^7$ |
| SMC | $5.935 \pm 0.228$ | $0.372 \pm 0.006$ | $0.425 \pm 0.003$ | $0.969 \pm 0.078$ | $1.002 \pm 0.072$ | $0.874 \pm 0.016$ | $5 \times 10^7$ |

also compare the wall-clock time for inference on 30K samples for a trained PITA model as well as classical baselines, SMC and PT. The training time for PITA on ALDP is 564.3 minutes. We note that PITA is roughly an order of magnitude faster than PT and two orders of magnitude faster than SMC after amortization. Therefore, the wall time breakeven point is around 120k samples vs. SMC and 300k samples vs PT for PITA.

Table 5: Inference time for 30k samples on ALDP

|  | Time (min) |
|---|---|
| PITA | 4.7 |
| SMC | 139.2 |
| PT | 59.5 |

# F    Additional Results

In Table 6, we report the Effective Sample Size (ESS) during PITA inference when importance weights are accumulated without SMC-based resampling on the ALDP dataset. The values are computed using 30000 samples for each of the temperature annealing steps. Note that these results correspond to a purely importance sampling-based setting (AIS) and thus differ from the full SMC formulation employed in PITA. The relatively low ESS values with AIS underscore the necessity of resampling for maintaining particle diversity and numerical stability. Additionally, we visualize the ESS as a function of the integration time during inference in Fig. 10. The results indicate that most of the ESS variation occurs at the early stages $t < 0.5$ (i.e., at higher noise levels), after which the ESS stabilizes. These findings suggest possible avenues for future work, such as exploring drift modifications aimed at reducing the variance of the importance weights.

# G    Ablation Studies

To evaluate the impact of our design choices, we perform a series of ablation studies examining: (1) the effect of annealing to 300K using different temperature jump sizes, (2) the choice of the $\gamma_t$ schedule, and (3) the role of resampling and different loss components, as well as comparing

Table 6: Log effective sample size (ESS) values without resampling for each temperature annealing step calculated across 30k samples on ALDP

| $T_L$ to $T_S$ | ESS ($\times 10^{-4}$) |
|---|---|
| 1200K to 755.95K | 2.77 |
| 755.95K to 555.52K | 8.66 |
| 555.52K to 300K | 3.79 |

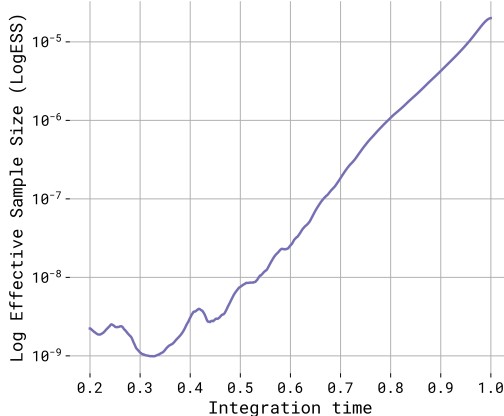

Figure 10: Log effective sample size (ESS) over integration time during inference.

out method with a simple classifier-free guidance approach. The temperature jump size ablation is performed for both ALDP and AL3, while the remaining studies are conducted on ALDP.

**Temperature Jump Sizes in Annealing**. To evaluate the effectiveness of the progressive annealing schedule, we compare the performance of models where the system is annealed from different starting temperatures to 300K. For ALDP, skipping intermediate temperatures has the most pronounced impact on energy distribution metrics, as shown in Table 7a and Fig. 12a. In the case of AL3, we find that sequential training is essential for reliably capturing all modes at the lower temperature, as illustrated in Fig. 11b. However, energy metrics degrade, likely due to small deviations in the sampled buffers at each annealing step, which accumulate over time. Accurately capturing the energy distribution in the sampled buffers at each intermediate temperature appears to be more difficult for AL3, which may contribute to the observed degradation. Nonetheless, capturing the correct modes remains a key priority, as modes lost during training are difficult to recover later. In contrast, mild deviations in the energy distribution can often be corrected through short MD relaxation steps, as demonstrated in Section 5.1.

$\gamma_t$ **schedule**. We analyze the effects of using different schedules for time-dependent $\gamma_t$ during inference. In particular, we annneal from 555.52K to 300.0K using a constant schedule, a linear schedule which linearly increases from $\gamma = 1$ to $\gamma = 1.85$, and a sigmoid schedule again, increasing from $\gamma = 1$ to $\gamma = 1.85$. Table 8 shows that the linear schedule generally performs best across the different evaluation metrics. It achieves the lowest values on the TICA metrics, while showing comparable performance to the sigmoid schedule on the energy-based metrics.

**Resampling, Energy Pinning Loss, and Classifier Free Guidance**. To evaluate the impact of resampling on sample quality, we perform inference from 556K to 300K without applying resampling. Across all metrics, we observe that resampling consistently improves performance. To assess the roles of the energy pinning loss and classifier-free guidance, we retrain a model for the 556K to 300K transition with each component removed. Omitting the energy pinning loss results in a slight improvement in TICA metrics but leads to a noticeable decline in energy metrics, indicating that the loss plays an important role in maintaining accurate energy distributions. Finally, we train a diffusion model on the data generated from PITA at 555.52K, and anneal to 300K simply by scaling the score by $\gamma$ (similar to classifier-free guidance approaches). This approach shows mixed results, offering no consistent improvement over MD-Diff (which is directly trained on samples at 300K) and performing below the level of PITA in all metrics.

Table 7: Effect of different starting temperatures on annealing performance for ALDP and AL3, evaluated at the final temperature of 300K.

(a) ALDP.

| $T_L$ to $T_S$ | Tica-$\mathcal{W}_1\downarrow$ | Tica-$\mathcal{W}_2\downarrow$ | Energy-$\mathcal{W}_1\downarrow$ | Energy-$\mathcal{W}_2\downarrow$ | $\mathbb{T}$-$\mathcal{W}_2\downarrow$ |
|---|---|---|---|---|---|
| 1200K to 300K | $\mathbf{0.100\pm0.004}$ | $\mathbf{0.297\pm0.019}$ | $6.438\pm0.024$ | $6.531\pm0.021$ | $0.301\pm0.023$ |
| 755.95K to 300K | $0.180\pm0.002$ | $0.611\pm0.003$ | $5.639\pm0.072$ | $5.683\pm0.070$ | $0.358\pm0.018$ |
| 555.52K to 300K | $0.121\pm0.004$ | $0.404\pm0.019$ | $\mathbf{1.541\pm0.009}$ | $\mathbf{1.619\pm0.010}$ | $\mathbf{0.270\pm0.023}$ |

(b) AL3.

| $T_L$ to $T_S$ | Tica-$\mathcal{W}_1\downarrow$ | Tica-$\mathcal{W}_2\downarrow$ | Energy-$\mathcal{W}_1\downarrow$ | Energy-$\mathcal{W}_2\downarrow$ | $\mathbb{T}$-$\mathcal{W}_2\downarrow$ |
|---|---|---|---|---|---|
| 1200K to 300K | $0.291\pm0.005$ | $0.558\pm0.003$ | $\mathbf{0.521\pm0.122}$ | $\mathbf{0.597\pm0.110}$ | $1.351\pm0.014$ |
| 755.95K to 300K | $0.234\pm0.009$ | $0.663\pm0.019$ | $17.147\pm0.105$ | $17.429\pm0.107$ | $0.751\pm0.006$ |
| 555.52K to 300K | $\mathbf{0.158\pm0.004}$ | $\mathbf{0.329\pm0.025}$ | $40.222\pm0.198$ | $40.978\pm0.208$ | $\mathbf{0.621\pm0.038}$ |

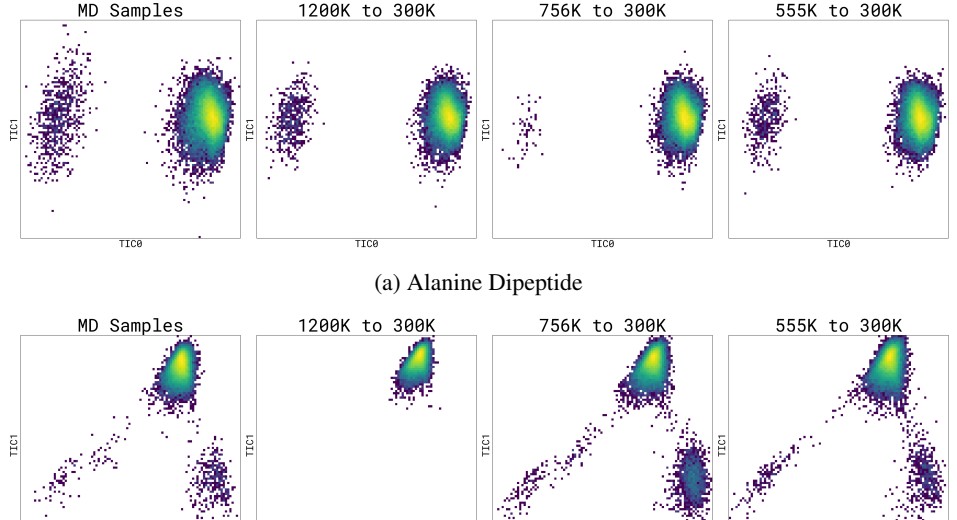

(a) Alanine Dipeptide

(b) Alanine Tripeptide

Figure 11: TICA plot of ALDP and AL3 samples obtained via annealing from various starting temperatures to 300K.

Table 8: $\gamma_t$ schedule ablation on ALDP.

| $\gamma$ Schedule | Tica-$\mathcal{W}_1\downarrow$ | Tica-$\mathcal{W}_2\downarrow$ | Energy-$\mathcal{W}_1\downarrow$ | Energy-$\mathcal{W}_2\downarrow$ | $\mathbb{T}$-$\mathcal{W}_2\downarrow$ |
|---|---|---|---|---|---|
| Constant | $0.115\pm0.005$ | $0.389\pm0.014$ | $1.485\pm0.100$ | $1.584\pm0.095$ | $\mathbf{0.258\pm0.030}$ |
| Linear | $\mathbf{0.095\pm0.009}$ | $\mathbf{0.243\pm0.048}$ | $1.453\pm0.099$ | $1.555\pm0.099$ | $0.275\pm0.058$ |
| Sigmoid | $0.113\pm0.009$ | $0.339\pm0.046$ | $\mathbf{1.443\pm0.087}$ | $\mathbf{1.550\pm0.087}$ | $0.345\pm0.027$ |

Table 9: Additional Ablation Results on ALDP.

| | Tica-$\mathcal{W}_1\downarrow$ | Tica-$\mathcal{W}_2\downarrow$ | Energy-$\mathcal{W}_1\downarrow$ | Energy-$\mathcal{W}_2\downarrow$ | $\mathbb{T}$-$\mathcal{W}_2\downarrow$ |
|---|---|---|---|---|---|
| PITA | $0.121\pm0.004$ | $0.404\pm0.019$ | $\mathbf{1.541\pm0.009}$ | $\mathbf{1.619\pm0.010}$ | $0.270\pm0.023$ |
| w/o resampling | $0.140\pm0.007$ | $0.452\pm0.027$ | $1.606\pm0.094$ | $1.676\pm0.075$ | $0.363\pm0.023$ |
| w/o energy pinning loss | $\mathbf{0.098\pm0.012}$ | $\mathbf{0.291\pm0.065}$ | $4.709\pm0.091$ | $4.722\pm0.090$ | $\mathbf{0.219\pm0.021}$ |
| MD-Diff + CFG | $0.137\pm0.007$ | $0.446\pm0.029$ | $8.106\pm0.025$ | $8.190\pm0.026$ | $0.383\pm0.042$ |

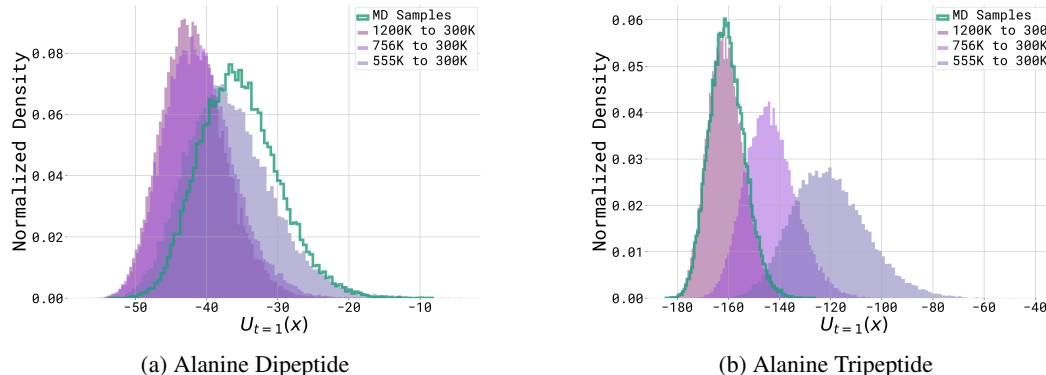

| (a) Alanine Dipeptide | (b) Alanine Tripeptide |

Figure 12: Energy distributions of ALDP and AL3 samples obtained via annealing from various starting temperatures to 300K. The model is trained on ground-truth MD samples at 1200K, and on annealed intermediate samples at 755.95K and 555.52K — the latter being our default training setting.

Table 10: Metrics across temperatures

| Temperature | Model | Tica-$\mathcal{W}_1 \downarrow$ | Tica-$\mathcal{W}_2 \downarrow$ | Energy-$\mathcal{W}_1 \downarrow$ | Energy-$\mathcal{W}_2 \downarrow$ | $\mathbb{T}$-$\mathcal{W}_2 \downarrow$ |
|---|---|---|---|---|---|---|
| 755.95K | PITA | **0.024 ± 0.004** | **0.125 ± 0.021** | 2.855 ± 0.083 | 2.886 ± 0.079 | **0.134 ± 0.013** |
| | TA-BG | 0.040 ± 0.002 | 0.178 ± 0.008 | **2.065 ± 0.044** | **2.140 ± 0.042** | 0.355 ± 0.007 |
| 555.52K | PITA | **0.141 ± 0.001** | **0.836 ± 0.004** | **1.420 ± 0.030** | **1.430 ± 0.027** | **0.219 ± 0.015** |
| | TA-BG | 0.337 ± 0.009 | 0.967 ± 0.013 | 48.486 ± 0.042 | 56.897 ± 0.059 | 1.135 ± 0.004 |
| 300K | PITA | **0.112 ± 0.006** | **0.379 ± 0.028** | **1.530 ± 0.068** | **1.615 ± 0.053** | **0.270 ± 0.023** |
| | TA-BG | 0.219 ± 0.013 | 0.685 ± 0.034 | 83.457 ± 0.070 | 86.176 ± 0.104 | 0.979 ± 0.012 |

# H   Training Dynamics Across Temperatures

In this section, we analyze the performance of the models (PITA and TA-BG) across different temperatures during annealing toward the target temperature on ALDP. Table 10 presents quantitative metrics, demonstrating that PITA consistently achieves lower discrepancies across all temperatures. Additionally, Figure 13 shows the Ramachandran plots at temperatures, further illustrating the ability of the model to generate physically realistic samples that capture the temperature-dependent conformational landscape at each step of the annealing process. TA-BG demonstrates reasonable performance at 755.95K when initialized with ground-truth samples, reflecting its ability to model high-temperature distributions under ideal conditions. However, its performance deteriorates when transitioning to lower temperatures using recursively generated samples for importance sampling, indicated by the mode collapse in the Ramachandran plots, where the conformational diversity sharply diminishes.

# I   Additional Experimental Details

## I.1   Parameterization Details

**PITA**. For LJ-13, we use equal loss weights for energy pinning, denoising score matching, and EBM distillation. We use the noise schedule of Karras et al. (2022), with the following parameters: $\sigma_{\min} = 0.05$, $\sigma_{\max} = 80$ and $\rho = 7$. The model uses EGNN (Satorras et al., 2021) with approximately 90k parameters, consisting of three layers and a hidden dimension of 32. For ALDP and AL3, the energy pinning, denoising score matching, and EBM distillation components of the loss are weighted equally at 1.0, with an additional target score matching loss weighted at 0.01. We use the same noise schedule as the LJ-13 experiment, using a smaller $\sigma_{\min}$ of 0.01. We use DiT (Peebles and Xie, 2023) comprising six layers and six attention heads, with a hidden size of 192 and a total of roughly 12 million parameters. All models are trained with a learning rate of $1 \times 10^{-3}$ without any weight decay. For ALDP and AL3, we use Exponential Moving Average (EMA) with a decay rate of 0.999, updating every gradient step.

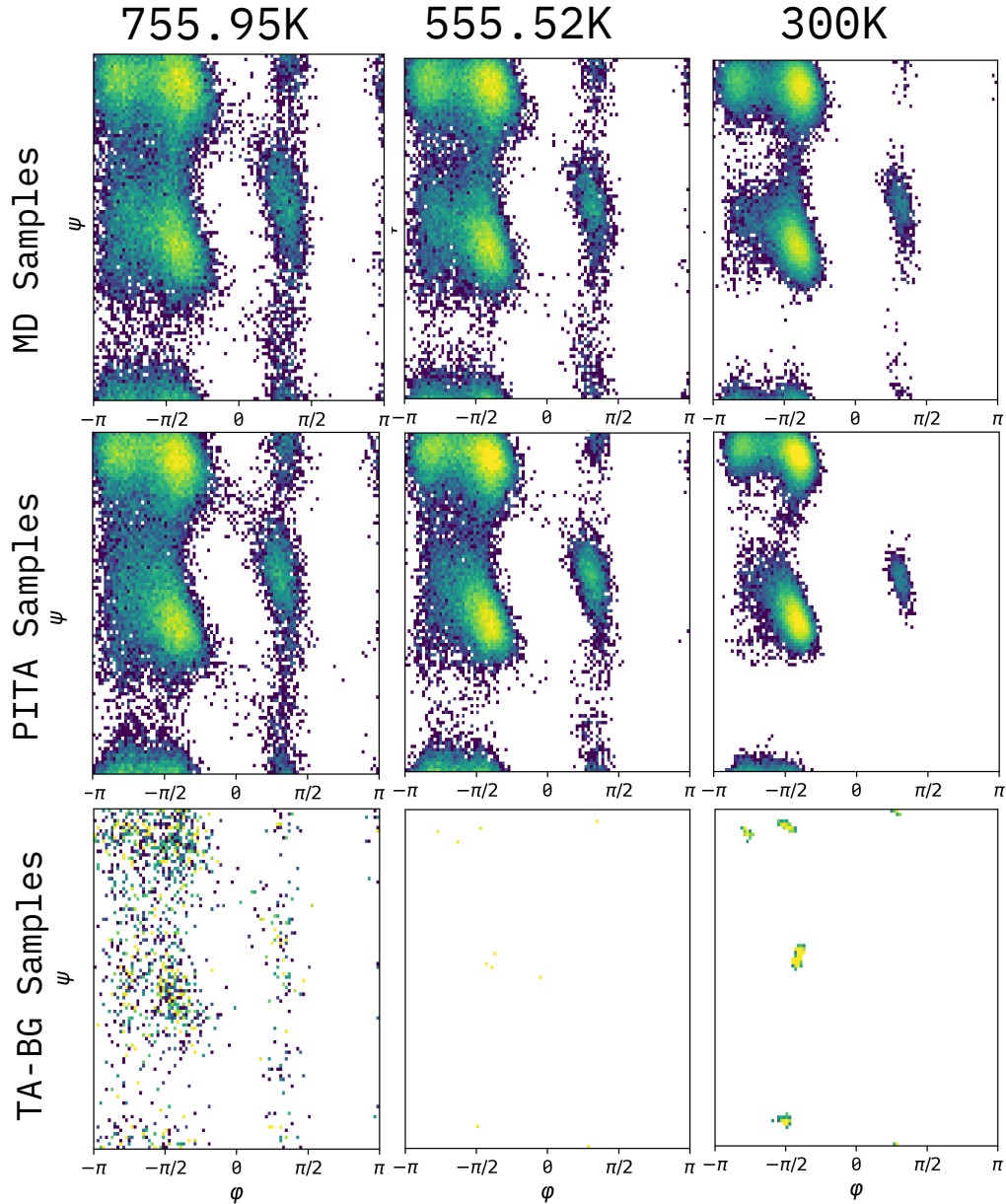

Figure 13: Ramachandran plots for Alanine Dipeptide (ALDP) obtained from different temperatures using 30k samples. We compare the samples from PITA and TA-BG with the ground-truth MD samples.

**MD-Diff**. We train the diffusion model on the MD trajectories generated at the target temperature. This serves as a strong baseline, since we have direct access to the ground-truth samples, unlike PITA and TA-BG. To ensure a controlled comparison, the length of the MD chain used to train the diffusion model is chosen such that the total number of energy evaluations matches the computational budget used to train PITA over all annealing steps. We provide further analysis on the mixing properties of different lengths of MD chains at low and high temperatures in Appendix D. We use a $\sigma_{\min}$ value of 0.005, while keeping the rest of the model hyperparameters the same as PITA.

**TA-BG**. In Schopmans and Friederich (2025), TA-BG trains a normalizing flow by minimizing the reverse Kullback–Leibler (KL) divergence at high temperature and progressively refining the model via importance sampling as the temperature is annealed toward the target distribution. We carefully adapt their training pipeline to ensure a consistent and fair comparison. Specifically, we initialize the training with ground-truth MD data rather than learned high-temperature samples, represent molecular configurations in Cartesian rather than internal coordinates, and use the same temperature annealing schedule as PITA. We use TarFlow (Zhai et al., 2025), configured with four meta blocks, each containing four attention layers and a hidden size of 256, resulting in approximately 12 million parameters. We use a learning rate of $1 \times 10^{-4}$ and employ 60,000 samples at the end of training for each temperature to compute the importance weights used in generating the buffer for the next temperature.

**MD-NF**. Similarly to MD-Diff, we train the normalizing flow model on the MD trajectories generated at the target temperature. We use TarFlow with the same model hyperparameters that we used for training TA-BG.

**Score Scaling**. We use a Score Scaling baseline which is a simple (but biased) modification to the score to attempt to sample from a different temperature. Specifically, given a score function $s_t(x_t)$ where we would normally sample with the SDE

$$dx_t = \left(-a_t x_t + \zeta_t^2 \frac{1+\xi_t}{2} s_t(x_t)\right) dt + \zeta_t \sqrt{\xi_t} dW_t, \quad x_0 \sim q_{t=0}(x) \tag{56}$$

we instead sample with

$$dx_t = \left(-a_t x_t + \gamma \zeta_t^2 \frac{1+\xi_t}{2} s_t(x_t)\right) dt + \zeta_t \sqrt{\xi_t} dW_t, \quad x_0 \sim q_{t=0}(x) \tag{57}$$

## I.2  Metrics

We evaluate model performance using both sample-based metrics and metrics that assess energy distributions. To compare energy distributions between generated samples and ground-truth molecular dynamics (MD) samples, we compute the 1D 1-Wasserstein and 2-Wasserstein distances on the energy histograms. For sample-based evaluation, we measure the 2D wrapped 2-Wasserstein distance of the internal dihedral angles, $\phi$ and $\psi$ (denoted as $\mathbb{T}$-$\mathcal{W}_2$). Additionally, we calculate the 2D 1-Wasserstein and 2-Wasserstein distances between the first two TICA axes of the ground-truth and generated samples.

## I.3  MD Parameters

**LJ-13 Parameters**. The Lennard-Jones (LJ) potential is an intermolecular potential that models interactions of non-bonding particles. The energy is a function of the interatomic distance of the particles:

$$\mathcal{E}^{\mathrm{LJ}}(x) = \frac{\varepsilon}{2\tau} \sum_{ij} \left( \left(\frac{r_m}{d_{ij}}\right)^6 - \left(\frac{r_m}{d_{ij}}\right)^{12} \right) \tag{58}$$

where the distance between two particles $i$ and $j$ is $d_{ij} = \|x_i - x_j\|_2$ and $r_m$, $c$, $\epsilon$ and $c_{osc}$ are physical constants. As in Köhler et al. (2020), we also add a harmonic potential to the energy so that $\mathcal{E}^{LJ-system} = \mathcal{E}^{\mathrm{LJ}}(x) + c_{osc}\mathcal{E}^{\mathrm{osc}}(x)$ This harmonic potential is given by:

$$\mathcal{E}^{\mathrm{osc}}(x) = \frac{1}{2} \sum_i \|x_i - x_{\mathrm{COM}}\|^2 \tag{59}$$

where $x_{\mathrm{COM}}$ is the center of mass of the system. We use $r_m = 1$, $c = 1$, $\varepsilon = 2.0$ and $c = 1.0$. For the LJ-13 dataset, we draw MCMC chains using the No-U-Turn-Sampler (NUTS) (Hoffman and Gelman, 2014)

**Alanine Parameters**. For MD data on ALDP and AL,3, we run two chains one for training and one for test. We use the same simulation parameters for both. For training data we sample shorter chains more frequently (every 100 md steps). To conserve disk space for long test chains, we save every 10k steps. Further parameters can be found in Table 11 and Table 12.

Table 11: `OpenMM` simulation parameters.

| | |
|---|---|
| Force field | `amber-14` |
| Integration time step | 1 fs |
| Friction coefficient | $0.3\,\mathrm{ps}^{-1}$ |
| Temperature | 300 K |
| Nonbonded method | `CutoffNonPeriodic` |
| Nonbonded cutoff | 2 nm |
| Integrator | `LangevinMiddleIntegrator` |

Table 12: Training and evaluation dataset parameters.

| | Train | Test |
|---|---|---|
| Burn-in period | 50 ps | 50 ps |
| Sampling interval | 0.1 ps | 10 ps |
| Simulation time | 50 ns | 1 μs |

## J  Pseudocode

In this section we provide Python pseudocode for easy of understanding and reimplementation.

```python
def resampled_inference(x0, T, score_model, energy_model, gamma, a,
    zeta, xi):
    xt = x0
    dt = 1 / T
    for t in linspace(0,1,T+1)[:-1]:
        # Define variables
        st = score_model(xt, t)
        ut = energy_model(xt, t)
        grad_Ut = grad(ut, xt)
        dUt_dt = grad(ut, t)
        # Equation (11)
        drift = (-a(t) * xt) + zeta(t)**2 / 2 * (st - gamma * xi(t) *
    grad_Ut)
        diffusion = zeta(t) * sqrt(xi(t)) * randn_like(xt)
        xt += drift * dt + diffusion * sqrt(dt)
        # Equation (12)
        dA = div(xt) * zeta(t) ** 2/ 2 - gamma * grad_Ut * ((-a(t) *
    xt) + zeta(t) ** 2 / 2 * st) - gamma * dU_dt
        At = dA * dt
        # Resample xt proportional to At with quasi monte carlo
        xt = resample(xt, At)
    return xt
```

Listing 1: Python implementation of resampled inference.

## K  Extended Related Work

**Annealed Importance Sampling**. In the context of AIS (Jarzynski, 1997; Neal, 2001), SMC samplers (Del Moral et al., 2006) and parallel tempering (Swendsen and Wang, 1986), our method reduces the number of energy evaluations by learning the models of intermediate marginals. Indeed, when the buffer of samples from the current temperature is sampled, training of the diffusion model does not require new energy evaluations (note that the gradients for target score matching can be

cached). Thus, the only time we need to evaluate the energies is for the importance sampling at the final step of the inference-time annealing and for the collection of samples via MCMC at a high temperature. Obviously, for sampling from the target density $\pi(x)$, the trained diffusion model, unlike AIS, allows producing uncorrelated samples without restarting the chain from the prior distribution.

