# OpenReview forum: "Progressive Inference-Time Annealing of Diffusion Models for Sampling from Boltzmann Densities"
_NeurIPS.cc/2025/Conference — NeurIPS 2025 spotlight_

### Official Review · Reviewer_SiSU · 2025-06-26

**Clarity:** 2
**Significance:** 3
**Originality:** 4
**Rating:** 5
**Confidence:** 4

**Summary:**

The goal is to draw samples from a distribution knowing its unnormalized density $\pi(x)$ — the authors design a way to additionally get approximate samples to make the sampling easier. The idea is to sequentially draw samples from tempered distributions $\pi(x)^{\beta}$, increasing the inverse temperature $\beta$ from $0$ (excluded) to $1$, as do classical Sequential Monte Carlo (SMC) algorithms.

**Repeat** until $\beta = 1$:

1. Increase $\beta$.

2. Use the unnormalized density $\pi(x)^\beta$ to get approximate samples.

3. Use the unnormalized density $\pi(x)^\beta$ and approximate samples to get better samples. To do so, train a diffusion model to learn and simulate energies of $\pi_t(x)^{\beta}$ where $t$ indexes the diffusion.

SMC algorithms typically do Steps 1 and 2. The authors' algorithm distinguishes itself by adding Step 3.

Moreover, the authors' implementation of Step 2 is also more sophisticated than SMC. The first time, step 2 is achieved by running an MCMC: it should converge relatively fast given that the multi-modality is flattened away. The rest of the time, step 2 is achieved by running an SMC (or FK-PDE) on the diffusion path at the previous temperature, raised to a power.

**Questions:**

Could the authors address the different points in the weaknesses section above?

**Ethical Concerns:**

["NO or VERY MINOR ethics concerns only"]

**Final Justification:**

I have raised my score to 5.

The reviewers had addressed my concerns with additional experiments. I have also read through other reviews and am satisfied with the clarifications the authors have provided.

**Limitations:**

Yes.

**Paper Formatting Concerns:**

No.

**Quality:**

3

**Strengths And Weaknesses:**

## Strengths

The authors' method is creative and interesting and seems to be improve over previous models on biological problems. It addresses many current questions:

- how to combine probability paths

The diffusion path avoids mass teleportation: it is usually simulated from samples.
The temperature path preserves the mode locations: it is usually simulated from an unnormalized density.

- how to obtain "good enough" samples from an unnormalized density in order to train a diffusion model to get "better" samples

This is an important challenge in the literature of diffusion samplers, as explained for example in *He et al., No Trick, No Treat: Pursuits and Challenges Towards Simulation-free Training of Neural Samplers*.

- how to use both combine samples *and* an unnormalized density to draw samples

## Weaknesses

**Dataset benchmarks**. It is encouraging to see sampling methods being used to make progress on real-world datasets. However, given the relevance and centrality of the authors' method to the sampling literature, it would be useful to include at least one experiment on a synthetic dataset such as a mixture of 50 Gaussians in 2 dimensions. This is because the sampling community as a whole does not have technical expertise in biological problems to judge the experiments section, because other competing methods are often benchmarked on a toy dataset, and also because it provides intuition on the performance of the method.

**Comparison with SMC in the experiments**. The most natural competitor to the author's PITA algorithm is a basic SMC that follows a temperature-annealing path. I appreciate the fact that some of the baselines (e.g. Temperature Annealed Boltzmann Generators) are very similar to SMC, but a vanilla SMC with a Langevin Kernel and a classical schedule for choosing the intermediate temperatures would be a convincing baseline in the experiments.

**Ablation study.**

Q1. For inference-time annealing, is it really helpful to do run an SMC through the diffusion path from the previous temperature, raised to a certain power? Can we not just directly reweigh the samples from $\pi^\beta$ to $\pi^{\beta + \Delta \beta}$ and then learn a diffusion model?

Q2. What is the difference in performance if PITA was run only with inference-time annealing and no model training?

**Comparison with SMC in the text**. Again, the most natural counterpart to the author's PITA algorithm is a basic SMC that follows a temperature-annealing path. It is also an algorithm most people in sampling are familiar path. Discussing how PITA and vanilla SMC differ would be very helpful to understanding PITA, as in the summary of this review.

**Lack of clarity**. I am quite familair with the topic and it took me many reads to understand what the algorithm what doing. Some things that could be more clearly explained:
- at a given temperature $\beta$, there are two probability paths that are being used: first, the diffusion path from the previous temperature, raised to a new power ("`inference-time annealing"), and then the diffusion path at the new temperature ("model training").
- why model training is useful in addition to inference-time annealing?
- Figure 1 is unclear to me. The two taxes of diffusion and temperature are very helpful. But the arrows are very confusing. I did not understand from the arrows that two probability paths are being used at a given temperature: the old diffusion path raised to a power ("inference-time annealing") and the new diffusion path ("model training").

**Unclear statements**.

Lines 175-177: "the DSM objective is not sufficient for training a good score model close to the target distribution [...] it has no information about the target distribution". This is not very clear: near the target distribution, the score model in Eq 13 is trained using samples $x_t$ that are almost from the target distribution. How is this compatible with the statement: "it has no information about the target distribution"?

Lines 61-62: "temperature annealing mixes modes by lowering high energy barriers, while diffusion paths avoid mass teleportation." This is a strong statement which would benefit from citing a few references. While it may be clear to experts, it is not obvious to any reader. For example, if one has not done the derivations in closed-form of the diffusion path from a Gaussian to a mixture of Gaussians, it is not obvious that mass is not teleported.

---

> ### Author Rebuttal · Authors · 2025-07-30
>
> We thank the reviewer for the detailed and constructive feedback. We greatly appreciate the positive evaluation of our work. In particular, the reviewer finds the proposed method "creative", "interesting", "addressing many current questions" and addressing "important challenges". In what follows, we address the concerns raised and provide additional evaluation requested by the reviewer.
>
> ## Dataset benchmarks
> > **Dataset benchmarks.** It would be useful to include at least one experiment on a synthetic dataset such as a mixture of 50 Gaussians in 2 dimensions.
>
> We thank the reviewer for their suggestion. We now include in this rebuttal the comparison on the synthetic Mixture of Gaussians dataset, training at $T_L=4$ and annealing to $T_S=1$. The comparison is presented in the table below, where we take the results for baselines from [1] and add the performance of PITA. Notably, PITA significantly outperforms all the considered baselines.
>
> **Table R3: Performance of methods on the mixture of 40 Gaussians in 2 dimensions.**
> |Method|Distance-W₂|Energy-W₂|Total Variance|
> |-------|------------|----------|--------------|
> |DDS|15.04±2.97|305.13±186.06|0.96±0.01|
> |PIS|6.58±1.68|79.86±7.79|0.95±0.01|
> |FAB|9.08±1.41|47.60±7.24|0.79±0.07|
> |iDEM|8.21±5.43|60.49±70.12|0.82±0.03|
> |**PITA**|**2.64±0.2**|**0.42±0.04**|**0.67±0.001**|
>
> ## Comparison with SMC in the experiments.
>
> We thank the reviewer for suggesting the inclusion of a vanilla SMC baseline with a Langevin kernel and a classical annealing schedule. We have incorporated this baseline in our experiments, and the results are reported in Table R1 (see our response to Reviewer JqJ5).
>
> > Comparison with SMC in the text. Again, the most natural counterpart to the author's PITA algorithm is a basic SMC that follows a temperature-annealing path. It is also an algorithm most people in sampling are familiar path. Discussing how PITA and vanilla SMC differ would be very helpful to understanding PITA, as in the summary of this review.
>
> Thank you for this constructive suggestion! We will add the corresponding discussion in the text.
>
> ## Questions about Ablations
> > For inference-time annealing, is it really helpful to do run an SMC through the diffusion path from the previous temperature, raised to a certain power? Can we not just directly reweigh the samples from $\pi(x)^\beta$ to $\pi(x)^{\beta+ \Delta \beta}$ and then learn a diffusion model?
>
> Following the reviewer's suggestion, we report the performance of inference-time annealing with importance sampling from $\pi(x)^{\beta+ \Delta \beta}$ using $\pi(x)^\beta$ as a proposal distribution, in Table R2 of our response to Reviewer Ldmh.
>
> > What is the difference in performance if PITA was run only with inference-time annealing and no model training?
>
> The inference-time annealing procedure heavily relies on the parametric models and is designed to sample from the unnormalized density of the energy-based model, i.e. $q_t(x)\propto \exp(-\gamma U_t(x;\eta))$. Thus, sampling entirely without training would not produce a meaningful distribution. Although one could perform sampling without training on the samples from the intermediate annealed densities, this approach consistently underperforms (see ablation study in Appendix E, line 858).
>
> ## Questions about Clarity of presentation
> > Some things that could be more clearly explained: at a given temperature $\beta$, there are two probability paths that are being used: first, the diffusion path from the previous temperature, raised to a new power ("inference-time annealing"), and then the diffusion path at the new temperature ("model training").
>
> We thank the reviewer for their insightful suggestion. We agree with the reviewer that the proposed phrasing can help clarify the two probability paths used in the PITA algorithm. We will update the corresponding subsections of the paper to better emphasize this distinction.
>
> > why model training is useful in addition to inference-time annealing?
>
> The inference-time annealing procedure is tightly coupled with the parametric models and is designed to sample from the unnormalized density of the energy-based model, $q_t(x)\propto \exp(-\gamma U_t(x;\eta))$. Training ensures that the parametric models capture these distributions accurately, which in turn makes the inference-time annealing effective. Jumping too far in temperature with inference-time annealing reduces performance of the method due to increased variance of the importance sampling weights. Model training is useful to "reset" these weights at a new lower temperatue.
>
> > Figure 1 is unclear to me. The two taxes of diffusion and temperature are very helpful. But the arrows are very confusing. I did not understand from the arrows that two probability paths are being used at a given temperature: the old diffusion path raised to a power ("inference-time annealing") and the new diffusion path ("model training").
>
> We appreciate the reviewer's feedback. In the current version, the diffusion path is represented by the horizontal line, as the standard diffusion does not involve any change in temperature. In contrast, the inference-time annealing path changes the temperature of marginals, which is illustrated by the curved line leading to the marginal $\pi(x)^\beta$ at a higher value of $\beta$. We hope this clarifies the interpretation of Figure 1.
>
> > Lines 175-177: "the DSM objective is not sufficient for training a good score model close to the target distribution [...] it has no information about the target distribution". This is not very clear: near the target distribution, the score model in Eq 13 is trained using samples that are almost from the target distribution. How is this compatible with the statement: "it has no information about the target distribution"?
>
> As $t \to 1$, equation (13) becomes
>
> $\mathbb{E}_{x_t,x} \lambda (t) ||x-D_t(x_t;\theta)||^2$
>
> $\to \mathbb{E}_{x_t,x}\lambda(t)|| \sigma_t^2 s_t(x;\theta)||^2 $
>
> because $\sigma_t \to 0$ and $x_t \to x$. This leads to unstable training of the score function $s_{t\approx 0}(x;\theta)$ with a finite dataset, as it converges to modeling the score of a uniform mixture of Gaussians with tiny variance around the dataset. This limitation of the DSM objective is well known, and we follow Target Score Matching [2] to address it. We will clarify this point in the text.
>
> > Lines 61-62: "temperature annealing mixes modes by lowering high energy barriers, while diffusion paths avoid mass teleportation." This is a strong statement which would benefit from citing a few references. While it may be clear to experts, it is not obvious to any reader. For example, if one has not done the derivations in closed-form of the diffusion path from a Gaussian to a mixture of Gaussians, it is not obvious that mass is not teleported.
>
> This is a valuable suggestion. We will add the discussion of the original work on diffusion models with SDEs [3], which proposed the reverse-time SDE and probability-flow ODE, demonstrating that diffusion paths can be simulated without probability mass teleportation. As well as [4] which nicely contrasts temperature and diffusion interpolations visually.
>
> ## Concluding remarks
>
> We thank the reviewer again for the thorough review and valuable feedback which strengthens the clarity and results of our paper through new ablations. We hope that our rebuttal definitely clarifies the all the great points raised in the review, and we politely invite the reviewer to ask any further questions or potentially endorse this paper more strongly if they are satisfied.
>
>
> ## References
> [1] OuYang, RuiKang, Bo Qiang, and José Miguel Hernández-Lobato. "Bnem: A boltzmann sampler based on bootstrapped noised energy matching." arXiv preprint arXiv:2409.09787 (2024).
>
> [2] De Bortoli, Valentin, Michael Hutchinson, Peter Wirnsberger, and Arnaud Doucet. "Target score matching." arXiv preprint arXiv:2402.08667 (2024).
>
> [3] Song, Yang, Jascha Sohl-Dickstein, Diederik P. Kingma, Abhishek Kumar, Stefano Ermon, and Ben Poole. "Score-based generative modeling through stochastic differential equations." arXiv preprint arXiv:2011.13456 (2020).
>
> [4] Máté, Bálint, and François Fleuret. "Learning interpolations between boltzmann densities." arXiv preprint TMLR (2023).

---

> > ### Comment · Reviewer_SiSU · 2025-08-03
> > **Answer to authors**
> >
> > Thank you authors for your response which addresses my questions.
> >
> > **Very related work**
> >
> > I have realized that the authors' method is in fact very close to another work [1] that has been published. While I understand that these projects may have been independently conducted, it is important to cite and discuss [1] as it has already been published, and also explain in what ways the authors' work differs from [1].
> >
> > [1] Rissanen et al., Progressive Tempering Sampler with Diffusion, ICML 2025

---

> > > ### Author Response · Authors · 2025-08-04
> > >
> > > We thank the reviewer for their participation during this rebuttal period. We are heartened to hear that our rebuttal was successful in addressing the reviewer's questions.
> > >
> > > We also thank the reviewer for flagging this related concurrent work. As the reviewer correctly points out, Rissanen et. al 2025 [1] was published at ICML 2025 this year. We note that the first public version of [1] was posted on arXiv on 5 June 2025---after the NeurIPS submission deadline.
> > >
> > > Nevertheless, we still agree that it is important to cite and discuss this concurrent work. In particular, while conceptually similar to PITA in that it considers annealed diffusion, our work has a few key benefits as compared to [1] that we now briefly outline.
> > >
> > > * **PITA benefits from annealed importance sampling (AIS)** over diffusion time, whereas **their sampling procedure is restricted to single-step importance sampling (IS).** Doing AIS as opposed to IS allows PITA to truly leverage the smoothing effects of the diffusion process, which drastically lowers the variance of the importance weights
> > > * **PITA's sampling is consistent where their sampling is biased** as their likelihoods are biased and have high variance as they use Hutchinson's trace estimator and a fixed step-size solver, which is known to be biased in these settings [2 Fig 6c].
> > > * While their evaluation is not directly comparable to ours, by visual inspection of Figs. 10 and 11 in [1], **their model is not able to capture the distribution of AL2 as well as PITA**, although uses ~1/2 of the energy evaluations. AL3 is not attempted in [1].
> > >
> > > We thank the reviewer again for highlighting this concurrent related work. We will further update the paper with an extended discussion of [1] and PITA in the related work section of the main paper. If the reviewer is satisfied with all of our answers as well as the discussion above, we would be encouraged if they considered upgrading their score. We are also happy to continue the discussion if there are any lingering doubts. Please do let us know.
> > >
> > > ## References
> > >
> > > [1] Rissanen et al., Progressive Tempering Sampler with Diffusion, ICML 2025
> > >
> > > [2] Klein et al., Equivariant Flow Matching, NeurIPS 2023

---

> ### Comment · Reviewer_SiSU · 2025-08-05
> **Answer to authors**
>
> Thank you authors.
>
> I have one remaining concern about the experiment "Performance of methods on the mixture of 40 Gaussians in 2 dimensions".
>
> Sometimes, the particles are initialized using a Gaussian distribution with a large enough covariance to "cover" all the modes. But this assumes prior knowledge on where the modes are located and makes the "search" problem easier.
>
> Could the authors run the 40 Gaussians experiments, and initialize the particles with the density of one of the modes only (so as to not assume that the user knows where the other modes are). This should make the mode "search" problem harder and fairer and would better illustrate whether PITA better finds modes compared to DDS, PIS, FAB, and iDEM.

---

> > ### Author Response · Authors · 2025-08-05
> >
> > We thank the reviewer for their continued participation in this rebuttal process. We now answer their newly posed question.
> >
> >
> > ## Initialization of samples in 2D GMM
> >
> > We acknowledge the reviewers comment about initializing the particles in one mode of the 40 GMM toy experiment. We would like to clarify that **PITA is not initialized with full mode coverage.** PITA is randomly initialized using a single sample $x_0 \sim \mathcal{N}(0,1)^2$ which is used to seed an MCMC chain to collect high temperature data. Note that the 40 GMM spans -50 to 50 in coordinate space. Consequently, PITA starts with knowledge of at most one mode.
> >
> > PITA gains knowledge of other modes outside of its initialized buffer by leveraging MCMC sampling at high temperature. The mode-mixing property of high-temperature distributions is a core motivation for PITA, and we leverage this behavior consistently across all our experiments. The reviewer may recall that this starting point is also shared in the ICML 2025 concurrent paper of Rissanen et. al 2025 which they flagged to us.
> >
> > In general our experiments demonstrate the capabilities of sampling from larger scale and more challenging sampling problems than those considered in DDS, PIS, FAB, and iDEM as we consider ALDP and AL3 in Cartesian coordinates. In this rebuttal we further discussed the infeasibility of iDEM to scale to our setup (please see Rebuttal Response to Reviewer 9xad). Furthermore, we demonstrated that PITA is capable of scaling with fewer energy evaluations of the target distribution. Thus, the sum total of our empirical findings suggest that PITA is more performant than past diffusion based samplers based on more interesting and challenging Boltzmann distributions.
> >
> > ## Concluding remarks
> >
> > In summary, PITA leverages the enhanced exploration capabilities of MCMC / MD at high temperatures to address the sampling challenge from a different and principled direction. This design is not only effective for synthetic benchmarks like the GMM, but also well-motivated by the molecular systems that are our primary application domain.
> >
> >
> > We hope this clarifies our experimental setup and addresses the very natural question raised by the reviewer. We thank them again for their great questions, and politely invite them to consider potentially upgrading their score if they feel satisfied with our responses. We are also happy to answer any further questions.

---

> > > ### Author Response · Authors · 2025-08-08
> > >
> > > We thank the reviewer again for providing their valuable feedback and engaging with us during the rebuttal period. As the end of the discussion period is fast approaching we were eager to understand if our most recent response was successful in alleviating the reviewers comments on the proposed 2D GMM experiments. We are more than happy to answer any final lingering doubts that might enable the reviewer to more enthusiastically endorse our paper. Please do let us know!

---

### Official Review · Reviewer_9xad · 2025-06-27

**Clarity:** 4
**Significance:** 2
**Originality:** 3
**Rating:** 5
**Confidence:** 4

**Summary:**

The paper proposes a scheme for training a diffusion sampler of a molecular Boltzmann distribution given access to the energy function and a set of high-temperature samples. In the scheme, diffusion models trained at the kth temperature are used to importance sample the target distribution at the (k+1)th temperature. These samples are used to train the next diffusion model. The process repeats until the desired temperature is reached.

**Questions:**

See above.

**Ethical Concerns:**

["NO or VERY MINOR ethics concerns only"]

**Final Justification:**

The authors have presented an approach to a longstanding problem that meaningfully advances current capabilities. Concerns about clarity were resolved in the rebuttal stage.

**Limitations:**

Yes

**Quality:**

3

**Strengths And Weaknesses:**

**Strengths**

The core idea proposed in the paper - that of bootstrapping a sampler by learning a sequence of distributions (using diffusion models) that anneal towards the target - is novel and conceptually solid. The formulation is rigorous and the connections with existing and prior literature are laid out well. The paper is overall well written and the supplementary experiments are detailed.

**Concerns or questions**

* The paper makes repeated claims about using less energy evaluations than previous methods, but these comparisons are not shown (on alanine peptides). Indeed, rather few prior methods are mentioned - standard diffusion sample baselines (DDS, iDEM, etc), while conceivably worse, are not shown in the results.

* As in persistent in this field, evaluations are made in settings of questionable practical utility - the small system size conceals problems that may make applications to larger systems impossible. In particular (1) poor coverage of dominant modes at high temperatures, e.g. phase transitions; (2) increasing number of intermediate temperatures required for effective importance sampling; (3) divergence computations to calculate the particle weights (4) distillation of the energy based models.

* Also as in persistent in this field, comparisons are made only on the basis of energy evaluations rather than wall-clock time.

* Why is it necessary to distill the score model into an energy based model? It should be possible to importance sample the target temperature with path RNDs computed only using the SDE drift terms. I would have guessed the weights at intermediate times, enabled by an explicit U_t, allow for frequent resampling using SMC - but the authors don't seem to do this.

* Authors should at least mention the vast body of prior literature on temperature replica exchange simulations.

* Authors notably omit ESS metrics in final comparisons, despite the key role of importance sampling in the training method.

* The development of a robust diffusion-based importance sampler of molecular systems in Cartesian coordinates is of independent interest and has not been reported in prior comparable work (iDEM, adjoint sampling). Authors should elaborate on the details and performance of the importance sampling weights in inference time annealing

* What is the meaning of the number of energy evaluations for the MD-diff baseline?

* To best contextualize the performance of method, baselines of normalizing flows trained on MD trajectories (i.e., SBG) should be shown.

---

> ### Author Rebuttal · Authors · 2025-07-30
>
> We thank the reviewer for their time and effort in assessing our paper. We are pleased to hear that the reviewer finds the key ideas in PITA of learning a sequence of diffusion models through annealing is "novel and conceptually solid". Moreover, we appreciate that the reviewer finds our formulation "rigorous" in conjunction with prior literature, and the overall paper "well written" with detailed supplemental experiments. We now turn to the main points of clarification raised by the reviewer below.
>
> ## Cost of energy evaluations in comparison to standard diffusion sampler baselines
>
> We acknowledge the reviewer's point that the high cost of energy function evaluations for standard diffusion sampler baselines, such as iDEM and DDS, is not provided in the current manuscript. We first note that the expensive energy evaluation cost of standard diffusion samplers like iDEM has been previously reported in prior works like "Adjoint Sampling" [1], Table 1, and in concurrent work "Adjoint Schrodinger Bridge Sampler" [2], Fig. 4.
>
> Moreover, the scalability of past diffusion samplers to molecular systems, in the **Cartesian coordinates** setting, remains challenging, if not downright infeasible. Since iDEM is simulation-free and more scalable than DDS, we bolster such folklore knowledge with private correspondence with the iDEM authors, who told us that iDEM failed to scale to ALDP even with substantial effort. We further attempted to reproduce these statements and trained iDEM on ALDP ourselves in Vacuum instead of implicit solvent, and observed that iDEM is unable to successfully sample in this easier setting with most modes missing and a poor energy distribution. We will update the manuscript with a discussion on this point and will provide new Rama plots for iDEM in the appendix.
>
> ## Evaluation on small peptides
>
> The reviewer makes an astute observation that the molecular systems considered in this paper are small peptides, and real-world application domains may demand conformation sampling at a much larger scale than considered here. We agree with the reviewer on this point. However, we would like to definitively clarify that prior to our work, PITA, **no other generative sampler** has been demonstrated in **Cartesian coordinates** for even ALDP, much less AL3. In addition, there is a burgeoning literature of combining generative models with classical MCMC techniques in an effort to "learn to sample" complex Boltzmann distributions. While preliminary, these methods promise the full benefit of deep learning, such as amortization of cost, faster than MD sampling of new points, and transferability to new, unseen molecular systems within a reasonable family. Consequently, in this literature, we position PITA as the current most scalable diffusion sampler, with expectations that future work---given the rapid progress of generative models for sampling---will enable scaling to more meaningful systems of interest. Thus, we believe PITA represents an inflection point wherein future work on generative samplers will be tested beyond even simpler synthetic energy functions such as 2D GMMs and Lennard-Jones potentials.
>
> We thank the reviewer for raising this point. We will include a discussion about the current limitations of diffusion samplers due to current scaling challenges in the conclusion.
>
> ## Further Questions
>
> > Also as in persistent in this field, comparisons are made only on the basis of energy evaluations rather than wall-clock time.
>
> We acknowledge the reviewer's comment regarding computational cost being tied to energy evaluation rather than simply wall clock time. In our Boltzmann sampling, the wall-clock time heavily depends on the quality of the energy approximation. For instance, using an energy function that takes into account quantum effects is orders of magnitude more expensive. Consequently, we report the number of energy evaluations in line with standard practice in the field.
>
> > Why is it necessary to distill the score model into an energy-based model? It should be possible to importance sample the target temperature with path RNDs computed only using the SDE drift terms. I would have guessed the weights at intermediate times, enabled by an explicit U_t, allow for frequent resampling using SMC - but the authors don't seem to do this.
>
> We appreciate the reviewer's nuanced comments. Parameterizing and learning the energy-based model is necessary for the importance sampling with the target density $\pi^\beta(x)$ after the inference-time annealing. Indeed, note that the inference-time annealing samples from the density $\propto\exp(-\gamma U_t(x;\eta))$, which does not perfectly match the target density. We discuss this in lines 156-161 and Appendix B.
>
> Furthermore, we perform an additional experiment using this method,  which we label as "FKC" in Table R2 of our response to Reviewer Ldmh. We note that this method is only correct if the learned scores perfectly align with $U_t$, which is difficult to achieve in practice.
>
> > Authors should at least mention the vast body of prior literature on temperature replica exchange simulations.
>
> Thank you for your comment. We will include further discussion on temperature replica exchange methods and their relation to our work on PITA.
>
> > Authors notably omit ESS metrics in final comparisons, despite the key role of importance sampling in the training method.
>
> Thank you for the note. ESS is not applicable here because our method uses SMC and resamples at every step. While ESS could be considered in settings without resampling, the performance differs drastically with and without resampling in our task. Therefore, we believe that reporting ESS for the AIS method would not accurately reflect the performance of our method as a whole, and we instead rely on other metrics. We will add a note to clarify this in the text.
>
> > The development of a robust diffusion-based importance sampler of molecular systems in Cartesian coordinates is of independent interest and has not been reported in prior comparable work (iDEM, adjoint sampling). Authors should elaborate on the details and performance of the importance sampling weights in inference time annealing.
>
> We thank the reviewer for this thoughtful suggestion and for recognizing the broader interest in this direction. We agree and will definitely elaborate on these points in the updated draft.
>
> > What is the meaning of the number of energy evaluations for the MD-diff baseline?
>
> For the MD-diff baseline, we run a Molecular Dynamics (MD) simulation with the predefined budget of energy evaluations to collect the training data at the target temperature and then train a diffusion model on the collected samples. The number of energy evaluations corresponds to the budget for the MD simulation.
>
> > To best contextualize the performance of the method, baselines of normalizing flows trained on MD trajectories (i.e., SBG) should be shown.
>
> We provide results of the normalizing flow trained on the MD trajectory data used for our MD-Diff baseline, which we call "MD-NF" [Table R2 Reviewer Ldmh]. We note that while this model can capture the energy distribution well, it fails to capture all the modes, as is evident from the TICA and Ramachandran (Rama) plots that will be included in the updated manuscript.
>
> ## Concluding remarks
>
> We thank the reviewer again for valuable comments and detailed feedback. We hope that our rebuttal addresses all of their concerns and encourages the reviewer to potentially consider updating their score if they so deem it. We are also available to answer any further questions that arise.
>
> ## References
>
> [1] Havens, Aaron, et al. "Adjoint sampling: Highly scalable diffusion samplers via adjoint matching." ICML (2025).
>
> [2] Liu, Guan-Horng, et al. "Adjoint Schrodinger Bridge Sampler." arXiv preprint arXiv:2506.22565 (2025).

---

> ### Comment · Reviewer_9xad · 2025-08-06
>
> I appreciate the authors' response. However, I find aspects of the response not fully satisfactory and would like to continue the discussion.
>
> > Q: Why is it necessary to distill the score model into an energy based model? It should be possible to importance sample the target temperature with path RNDs computed only using the SDE drift terms. I would have guessed the weights at intermediate times, enabled by an explicit U_t, allow for frequent resampling using SMC - but the authors don't seem to do this.
>
> > A: We appreciate the reviewer's nuanced comments. Parameterizing and learning the energy-based model is necessary for the importance sampling with the target density
>  after the inference-time annealing.
>
> First of all, this does not address my question because importance sampling against a target density is also possible with path RNDs that make use only of the SDE drift (do authors agree?). More curiously however it seems like the authors _do_ in fact resample as stated later in the rebuttal and (upon re-inspection) in the main text, though this is not stated prominently. This raises concerns that the method is not presented as clearly as it could be. To remediate this point I would ask that the authors produce pseudocode for the inference algorithm and agree to include it in the revision.
>
> > Q: Authors notably omit ESS metrics in final comparisons, despite the key role of importance sampling in the training method.
>
> > A: ESS is not applicable here because our method uses SMC and resamples at every step. While ESS could be considered in settings without resampling, the performance differs drastically with and without resampling in our task.
>
> I interpret this to mean that the ESS is poor without resampling. I would not be surprised, and would not hold it against the authors --- there are no diffusion samplers that obtain good ESS on these systems. However it is important to transparently show these metrics so we can have common knowledge of where to improve.

---

### Official Review · Reviewer_Ldmh · 2025-07-03

**Clarity:** 3
**Significance:** 2
**Originality:** 3
**Rating:** 5
**Confidence:** 3

**Summary:**

This paper proposes a novel method that combines annealing and diffusion-based samplers to sample from unnormalized target distributions. The key idea is to begin at a high-temperature distribution, where mixing is easier for standard MCMC methods, and progressively anneal toward the target distribution. At each annealing step, the approach alternates between two components: (i) learning a diffusion-based sampler via time-dependent score and energy models at that temperature, and (ii) generating samples using a Feynman-Kac PDE framework for the next, lower temperature.

In empirical study, the method is evaluated on a series of synthetic particle systems. The proposed method generally produces qualitatively and quantitatively better results than existing baselines. On the most challenging task, Alanine Tripeptide, the method achieves good mode coverage but underperforms on certain evaluation metrics.

**Questions:**

- If I understand correctly, the method involves training separate models at multiple temperature levels. Are these models trained independently, or is it amortized across temperatures? It would also be helpful to clarify how this overhead compares to other methods in terms of computational efficiency and scalability.

- I wonder why the authors chose to use two separate models for energy function and score (or equivalently, denoiser)? In principle the score function can be obtained by taking the gradient of the energy function. Is this separation primarily to avoid the computational and memory burden of backpropagation?

- Have the authors conducted an ablation study on the four types of losses used in Algorithm 1? Are all of them necessary? Do they receive equal weight in the final objective? Are they trained under the same optimizer settings?

- Have the authors considered a baseline where a diffusion model is trained at a high temperature, and then sampling is performed using the Feynman-Kac PDE framework at the target (lower) temperature?

- How do you tune the hyper-parameters for all methods?

- Do you have any guidance in setting the annealing schedules for new tasks?

- Do diffusion models trained at different temperatures require different optimization settings or hyperparameter tuning? If yes, do you have practical guidance on that?

-This paper [1] also explores diffusion-based sampling with access to the target-mode information and may serve as a useful point of comparison:

[1] LEARNED REFERENCE-BASED DIFFUSION SAMPLING
FOR MULTI-MODAL DISTRIBUTIONS
Maxence Noble∗, Louis Grenioux∗, Marylou Gabrie & Alain Oliviero Durmus

Minor and writing-related:

-  In this expression "U_t = -log p_t(x) + const", is the constant independent of t?

- Eq 10: Can you explain the right equation? What does the "\approx" mean and why it holds?

- Algorithm 1: the loss function names in the optimizer lines are inconsistent with those in previous lines

- Algorithm 1: It would be helpful to make the algorithm self-contained by specifying inputs such D_t and U_t.

- Table 2 and 3: does the reported #energy evals for the proposed method  include the evaluations used in the initial MCMC sampling phase? Could the authors provide the MCMC configuration details (e.g., number of steps, acceptance rates), and are the final results sensitive to these settings?

- Table 3: what is "PITA" vs "PITA (w/o relaxation)"?

- On page 5 line 164: "one subsection" instead of "two"?

**Ethical Concerns:**

["NO or VERY MINOR ethics concerns only"]

**Final Justification:**

Most of my questions are resolved.

While PITA incurs additional training overhead, it provides faster inference and has the potential to build transferrable samplers across systems.

**Limitations:**

One limitation acknowledged by the authors is the need to train both score and energy networks, which increases the training complexity

Additional imitations include

- The need to train these models at several temperature levels, further raising the training cost.

- Lack of comparison to some of the classical sampling approaches.

**Quality:**

3

**Strengths And Weaknesses:**

Strengths:

- The proposed method is intuitive and well-motivated, leveraging a progressive annealing path and coarse simulation data (by MCMC) to address challenges in existing diffusion-based methods.

- The proposed method empirically scales to systems where prior diffusion-based approaches struggle.

Weakness:

- Some parts of the writing could be improved—there are inconsistencies and instances of imprecise or vague explanations (see specific comments).

- The paper lacks empirical comparisons to classical baselines such as replica exchange or sequential Monte Carlo (with temperature annealing), making it difficult to assess the overall significance of the proposed approach.

- The proposed approach can be less flexible as it requires training diffusion models at different temperature, and relies on specifying the initial temperature and annealing schedule.

---

> ### Author Rebuttal · Authors · 2025-07-30
>
> We thank the reviewer for the detailed and constructive feedback. We appreciate the positive assessment of the proposed method. Namely, the fact that the reviewer finds it to be "intuitive", "well-motivated", and "addressing existing challenges in diffusion-based methods". In what follows, we address the concerns raised and answer their questions grouped by theme.
>
>
> ## Empirical comparison with classical baselines
>
> We acknowledge the reviewer concern that our paper could benefit from the addition of classical baselines(SMC and Replica Exchange/Parallel Tempering). We agree with this feedback and have included the suggested baselines in this rebuttal, as shown in Table R1 in our response to Reviewer JqJ5.
>
> ## Computational overhead and efficiency of training multiple diffusion models
>
> We appreciate the reviewer’s comment that the discussion of the computational overhead and efficiency of training separate models amortized across temperatures may not have been sufficiently clear in the manuscript. We now attempt to add increasing clarity to these points. We begin by first noting that in practice we find that sequential fine-tuning of the model demonstrates the best performance---a point thats outlined also in lines 199-201 of the main paper. More precisely, for the subsequent training of a diffusion model at a lower temperature $1/\beta_{i+1}$ we fine-tune the model trained at the previous temperature $1/\beta_{i}$. Notably, this strategy allows for reducing the number of required parameters (compared to amortization over temperatures) and the number of training iterations (compared to training from scratch for every temperature).
>
> Furthermore, we wish to highlight that by cost we include both the training time as well as the number of target energy function evaluations needed, which is an important cost associated to real world Boltzmann distributions for molecular systems. We will clarify these computational efficiency gains and their overhead in the updated draft of the main paper.
>
>  ## Questions
>
> > I wonder why the authors chose to use two separate models for energy function and score (or equivalently, denoiser)? In principle the score function can be obtained by taking the gradient of the energy function. Is this separation primarily to avoid the computational and memory burden of backpropagation?
>
> The reviewer makes an interesting observation. Indeed, the score function can be obtained as a gradient of the energy function. There are three reasons why we use the a separate score model.
> 1. As noted by the reviewer the computational burden is reduced, and in particular this scheme allows us to avoid the computation of the laplacian of the energy-based model for computing the weights.
> 2. Training separate score and energy models is the current SOTA in energy model training following [1]. We get the score function for free given our training scheme.
> 3. The score function in general produces better samples than the gradient of the energy function, therefore is a better proposal distribution in practice.
>
> We think this is a cool and important component of PITA and will add additional discussion in the text.
>
> > Have the authors conducted an ablation study on the four types of losses used in Algorithm 1? Are all of them necessary? Do they receive equal weight in the final objective? Are they trained under the same optimizer settings?
>
> The four losses in Algorithm 1 consist of two groups: 2 losses for training the score model and 2 losses for training the EBM. For training the score model when both training samples and the target unnormalized density are available, we follow the best practice in the field so far [2]. For training the energy-based model, we follow [1] and distill it from the score model. The novel loss introduced here is the "energy pinning" loss, which we ablate in Appendix E (see Table 6 for comparison). We hope this now sufficiently clarifies and the reviewers question.
>
> > Have the authors considered a baseline where a diffusion model is trained at a high temperature, and then sampling is performed using the Feynman-Kac PDE framework at the target (lower) temperature?
>
> We have updated our results below to include
> 1. Training a diffusion model and only rescaling the score to anneal from 1200K to 300K (ScoreScaling),
> 2. Using the Feynman-Kac Correctors [5] framework to anneal from 1200K to 300K (FKC).
>
> As can be seen in the table below, neither of these frameworks can successfully capture the lower temperature distribution on this task.
>
> **Table R2: Performance of methods for the ALDP sampling task.**  The starting temperature is $T_L$ = 1200 K, annealed to target $T_S$ = 300 K. Metrics are calculated over 10k samples and standard deviations over 3 seeds.
>
> |Method|Rama-KL|Tica-W₁↓|Tica-W₂↓|Energy-W₁↓|Energy-W₂↓|T-W₂↓|#EnergyEvals|
> |------|-------|--------|--------|-----------|-----------|------|-------------|
> |PITA|4.773±0.460|**0.112±0.006**|**0.379±0.028**|1.530±0.068|1.615±0.053|**0.270±0.023**|5×10⁷|
> |MD-Diff|**1.308±0.072**|0.113±0.001|0.579±0.004|3.627±0.023|3.704±0.026|0.310±0.001|5×10⁷|
> |MD-NF|13.533±0.024|0.138±0.003|0.586±0.003|**0.551±0.062**|**1.198±0.069**|0.403±0.045|5×10⁷|
> |TA-BG|14.993±0.002|0.219±0.013|0.685±0.034|83.457±0.070|86.176±0.104|0.979±0.012|5×10⁷|
> |FKC|14.392±0.909|0.217±0.000|0.649±0.001|11.281±0.025|11.466±0.027|2.120±0.024|5×10⁷|
> |ScoreScaling|4.588±0.467|0.183±0.002|0.608±0.008|10.282±0.020|10.460±0.019|0.550±0.036|5×10⁷|
>
> > How do you tune the hyper-parameters for all methods?
>
> For all methods we start with the parameters from SOTA methods in the literature then perform coordinate-wise grid searches over the most relevant parameters to our task using the best performing setting on a validation set at temperature 1200k. For loss weights we simply approximately balanced the magnitude of the losses on the training set, e.g. we used a weight of 0.01 for the target score matching loss to ensure it is on the same scale as the other losses.
>
> > Do you have any guidance in setting the annealing schedules for new tasks?
>
> For simplicity, we follow the standard practice in the literature and consider the geometric annealing schedule [3]. This, however, can be potentially improved by incorporating modern advanced schedule optimization techniques [4]. We leave exploration of annealing schedules, including dynamic or learned ones, as interesting directions for future work. In theory, the standard practice of annealing to match a target ESS should also work in our setting.
>
> > Do diffusion models trained at different temperatures require different optimization settings or hyperparameter tuning? If yes, do you have practical guidance on that?
>
> We find that our approach performs consistently well across different temperatures without additional hyperparameter tuning. Specifically, we keep the training settings (e.g., learning rate, number of iterations) constant across all temperature levels.
>
> > This paper also explores diffusion-based sampling with access to the target-mode information and may serve as a useful point of comparison: LEARNED REFERENCE-BASED DIFFUSION SAMPLING FOR MULTI-MODAL DISTRIBUTIONS Maxence Noble∗, Louis Grenioux∗, Marylou Gabrie & Alain Oliviero Durmus
>
> We thank the reviewer for pointing out the relevant piece of literature. We have added a discussion of this work into the related work section.
>
> ## Minor and writing-related comments
> We thank the reviewer for the detailed minor writing comments.
>
> > In this expression "U_t = -log p_t(x) + const",  is the constant independent of t?
>
> Yes, that is the log normalization constant.
>
> > Eq 10: Can you explain the right equation? What does the "\approx" mean and why it holds?
>
> It is approximate as $U_t$ is a learned model and $\pi$ is the true distribution.
>
> > Table 2 and 3: does the reported #energy evals for the proposed method include the evaluations used in the initial MCMC sampling phase? Could the authors provide the MCMC configuration details (e.g., number of steps, acceptance rates), and are the final results sensitive to these settings?
>
> Yes, the reported number of energy evaluations includes those in the initial sampling phase. Note that we collected samples using MCMC for LJ-13 and Molecular Dynamics for the peptide experiments. These configurations are provided in Appendix G.3. Since we use a fairly converged chain, the final results are not very sensitive to these configuration details.
>
>
> ## Concluding remarks
> We thank the reviewer again for their detailed feedback that helped us improve our work. We very politely invite the reviewer to increase their score if they feel our rebuttal was successful in answering all their raised concerns. We also remain available for further discussion and for addressing any lingering doubts. Please do let us know.
>
>
> ## References
> [1] Thornton, James, Louis Béthune, Ruixiang Zhang, Arwen Bradley, Preetum Nakkiran, and Shuangfei Zhai. "Composition and control with distilled energy diffusion models and sequential monte carlo." arXiv preprint arXiv:2502.12786 (2025).
>
> [2] De Bortoli, Valentin, Michael Hutchinson, Peter Wirnsberger, and Arnaud Doucet. "Target score matching." arXiv preprint arXiv:2402.08667 (2024).
>
> [3] Del Moral, Pierre, Arnaud Doucet, and Ajay Jasra. "Sequential monte carlo samplers." Journal of the Royal Statistical Society Series B: Statistical Methodology 68, no. 3 (2006): 411-436.
>
> [4] Syed, Saifuddin, Alexandre Bouchard-Côté, Kevin Chern, and Arnaud Doucet. "Optimised annealed sequential Monte Carlo samplers." arXiv preprint arXiv:2408.12057 (2024).
>
> [5] Skreta, Marta, et al. "Feynman-kac correctors in diffusion: Annealing, guidance, and product of experts." ICML (2025).
>
> [7] Karras, T., et al. "Elucidating the design space of diffusion-based generative models." NeurIPS (2022).

---

> > ### Comment · Reviewer_Ldmh · 2025-08-06
> >
> > I would like to thank the authors for their detailed rebuttal. Most of my questions have been properly addressed.
> >
> > However, I have a few follow-up questions regarding the complexity comparison to existing training-free methods and would appreciate further clarification on the practical advantages of the proposed method:
> >
> > In table R1,  it appears that there is no clear performance winner among PITA, PT, and SMC, given the same number of energy evaluations. In that case, could you also provide training/sampling wall-clock time comparison? I suspect that training-based samplers, including PITA, may incur higher overhead. Yet, a quantitative comparison would help assess the practical implications of these trade-offs more clearly. I would also encourage the authors to share their thoughts on potential advantages of PITA compared to these classical training-free sampling methods.

---

### Official Review · Reviewer_JqJ5 · 2025-07-06

**Clarity:** 3
**Significance:** 2
**Originality:** 2
**Rating:** 4
**Confidence:** 3

**Summary:**

This paper introduces Progressive Inference-Time Annealing (PITA), a generative framework that progressively trains a cascade of diffusion models to sample from low-temperature Boltzmann distributions. The core idea is to combine two interpolation strategies: (1) temperature annealing, which flattens energy barriers and enables better mode mixing; and (2) diffusion smoothing, which enables efficient resampling at inference time via Feynman–Kac-weighted reverse SDEs. By alternating between training a diffusion model at a given temperature and then resampling to a colder temperature using learned score and energy models, PITA can produce samples that align closely with the Boltzmann target at low temperatures.

**Questions:**

1. Since the method relies on both a score model and an energy model at each temperature level, is the computational and training cost substantially higher than standard diffusion models or Boltzmann generators?
2. The proposed method appears to be reminiscent of annealed Langevin dynamics and predictor-corrector samplers discussed in [9], which also gradually anneal the target distribution by adjusting the noise scale. Could the authors discuss the relationship and trade-offs? What are the advantages of your approach?
3. In [10], the authors proposed an energy model with adaptive temperature control by reparameterizing the energy function, enabling built-in temperature adjustment in the generative model. Could the authors comment on this approach and discuss the advantages of PITA compared to such adaptive-temperature energy models?

**References:**

[9] Score-based generative modeling through stochastic differential equations, ICLR 2021

[10] Learning energy-based models by cooperative diffusion recovery likelihood, ICLR 2024

**Ethical Concerns:**

["NO or VERY MINOR ethics concerns only"]

**Final Justification:**

Thank you for your comprehensive rebuttal. Your clarifications on the core novelty and the re-contextualization of your evaluation's scope have been effective. The new experiments provide evidence supporting your claims and successfully distinguish your work from prior work. You have addressed my concerns. I have raised my score to 4.

**Limitations:**

Yes

**Paper Formatting Concerns:**

No formatting concerns.

**Quality:**

3

**Strengths And Weaknesses:**

**Strengths**

The paper is well-written and presents a clean formulation of its algorithmic ideas. The integration of Feynman–Kac theory into diffusion inference is interesting and, to my knowledge, novel in the generative modeling literature. Moreover, the approach achieves significant gains in sample quality and coverage compared to prior Boltzmann generator and diffusion methods, especially in physically meaningful molecular domains.

**Weaknesses**

1. The core novelty of the two interpolants in this work (temperature annealing and diffusion smoothing) is not sufficiently distinguished from existing literature. It seems that both techniques have been widely studied in the MCMC, diffusion-based modeling or energy-based modeling literature.
2. Annealing has long been used in MCMC [1] and has been applied to diffusion models [2–4].
3. The idea of diffusion smoothing looks similar to annealed importance sampling [5] and was later extended to deep energy-based models [6–8].
4. From my understanding, the method reads like a combination of known components, and the authors may wish to clarify why this framework yields fundamentally new insights or performance advantages beyond prior annealed sampling or diffusion-EBM hybrids.
5. The scope of the evaluation is limited. While the authors demonstrate promising results on small physical systems like LJ-13 and alanine peptides, it is unclear whether this method generalizes to broader settings (e.g., image or text generation, or large protein folding datasets). The method is currently tailored for low-dimensional molecular systems where accurate energy evaluations are feasible.


**References:**

[1] Bayesian learning via stochastic gradient Langevin dynamics, ICML 2011

[2] Generative modeling by estimating gradients of the data distribution, NeurIPS 2019

[3] Denoising diffusion probabilistic models, NeurIPS 2020

[4] Improved techniques for training score-based generative models, 2020

[5] Annealed importance sampling, Statistics and computing, 2001

[6] Learning energy-based models by diffusion recovery likelihood, ICLR 2021

[7] A Langevin-like sampler for discrete distributions, ICML 2022

[8] Score-based diffusion meets annealed importance sampling, NeurIPS 2022

---

> ### Author Rebuttal · Authors · 2025-07-30
>
> We thank the reviewer for their valuable feedback and the time spent. We appreciate that the reviewer finds the paper to be "well-written" and "the integration of Feynman–Kac theory into diffusion inference to be interesting and novel". In what follows, we address the potential misunderstandings and concerns raised.
>
> ## Distinguishing the novelty of the two interpolants
> We acknowledge the reviewer's comment that the core novelty of our two introduced interpolants may not have been sufficiently distinguished from existing literature. We now clarify this important point further.
>
> > The core novelty of the two interpolants in this work [...] is not sufficiently distinguished from existing literature.
>
> > From my understanding, the method reads like a combination of known components, and the authors may wish to clarify why this framework yields fundamentally new insights or performance advantages beyond prior annealed sampling or diffusion-EBM hybrids.
>
> This framework yields performance advantages beyond prior annealed sampling or diffusion-EBM hybrids by being the first model to perform consistent simultaneous annealing in both the diffusion and temperature axes.
>
> One of the core novelties of this work is the Sequential Monte Carlo (SMC) scheme for the inference-time annealing of diffusion models, which enjoys both the efficiency of expressive diffusion models and the consistency of SMC estimators. **This combination of diffusion and temperature annealing in a consistent manner is what distinguishes PITA from the existing literature**. In particular, compared to the previous literature, this procedure has the following merits:
> 1. Unlike Langevin Monte Carlo [1], which converges only in the limit of an infinite number of steps, it provides a consistent estimator w.r.t. the number of simulated particles. This fact allows us to curate new lower temperature (consistent) datasets to train new diffusion models---a fact that is not possible with simple Langevin Monte Carlo without incurring bias.
> 2. The critical advantage of PITA is that by combining both interpolation strategies, we achieve more stable and performative training of the final diffusion sampler. This combination, to the best of our knowledge, is novel and in particular benefits from our inference time SMC scheme that is tailor-made for diffusion models. As we stress, in the main paper, no existing diffusion sampler is able to tackle the molecular energy systems we consider in Cartesian coordinates. This is because training such samplers without data is extremely challenging and numerically unstable due to the poor estimation of the Stein score, as well as finding the modes in a complex energy landscape to estimate the score in the first place.
>
> > Annealing has long been used in MCMC [1] and has been applied to diffusion models [2–4].
>
> The reviewer makes an astute observation that Annealing has been applied to diffusion models. **In new experiments, we demonstrate that PITA performs better than Annealing in MCMC, and existing annealed diffusion models**.
>
> 1. We first add two baselines of annealed MCMC with sequential Monte Carlo and parallel tempering (PT) strategies.  As it can be seen in Table R1 below, PITA performs better overall in capturing the distribution given the same budget of energy evaluations
>
> **Table R1**: Performance of the SMC and PT baselines on ALDP. For SMC, we take 10 annealing temperatures between 1200K and 300K using a geometric schedule, taking 166 steps per temperature with 30k particles. For PT, we use the same 4 annealing temperatures as PITA, running 25 MD steps between particle exchanges for a total of 50k iterations.
>
> | Method| Rama-KL ↓| Tica-W₁ ↓| Tica-W₂ ↓| Energy-W₁ ↓| Energy-W₂ ↓| T-W₂ ↓| #Energy Evals|
> |-|-|-|-|-|-|-|-|
> | PITA |**4.773±0.460**|**0.112±0.006**|**0.379±0.028**|1.530±0.068|1.615±0.053|**0.270±0.023**|5×10⁷|
> | PT|7.306±1.077|0.625±0.010|0.895±0.016|4.652±0.015|4.689±0.014|0.911±0.004| 5×10⁷|
> | SMC|5.935±0.228|0.372±0.006|0.425±0.003|**0.969±0.078**|**1.002±0.072**| 0.874±0.016|5×10⁷|
>
> 2. Second, we add a baseline of "score scaling", which is the typical strategy in classifier-free guidance approaches to approximately anneal diffusion models [Table R2 Reviewer Ldmh]. PITA substantially outperforms this method. We note that PITA is consistent where score scaling is not, which explains PITA's superior performance. Score scaling does not produce correct marginals both theoretically and empirically, as also demonstrated in the recent Feynmann-Kac Correctors paper [11], Section 5.2.
>
> > The idea of diffusion smoothing looks similar to annealed importance sampling [5] and was later extended to deep energy-based models [6–8].
>
> Compared to Annealed Importance Sampling (AIS) [5], it follows the diffusion path raised to the power $\beta$ (i.e. $p_t(x)^\beta$) instead of the geometric averaging (i.e. $\pi(x)^{t\beta}p_0(x)^{(1-t)}$). The use of diffusion paths is crucial for our setting as it reduces the variance of importance weights.
>
> In addition, we note that previous works on deep energy-based models [6-8] combine the standard inference with Langevin dynamics. This means that the inference procedure converges only in the limit of an infinite number of iterations. This is in stark contrast with the proposed consistent estimator used in PITA, which, as described above, allows for the consistent curation of lower temperature datasets.
>
> ## Scope of Evaluation
> We value the reviewer's concern that the scope of evaluation may initially appear limited. However, we would like to politely push back against this characterization. We begin by recalling that our paper is concerned with sampling from an unnormalized density---a problem domain that is well known to be significantly harder than conventional generative modelling. This is because, unlike generative modelling, we are afforded **no samples from the target distribution**, and yet we must draw i.i.d. samples from it. Due to this challenge, a majority of work has focused on smaller molecular systems in bespoke internal coordinates, which are of smaller dimensionality and are provided through exact domain knowledge for a given system. In contrast, energy functions in Cartesian coordinates are more challenging as they represent the **full dimensionality of the molecular system** and, as a result, are much larger. As a result, we wish to again reiterate this subtle but important point, existing diffusion samplers have failed to even scale to Alanine Tripeptide (AL3) with 99 dimensions, which our method, PITA, effectively tackles. Consequently, we argue that such an empirical demonstration goes beyond the current benchmarks in the literature.
>
> We thank the reviewer again for their comment, and we will further update the manuscript to highlight the challenge of sampling in Cartesian coordinates.
>
> ## Questions
> > Since the method relies on both a score model and an energy model at each temperature level, is the computational and training cost substantially higher than standard diffusion models or Boltzmann generators?
>
> Training a score model and an energy model incurs higher cost than a standard diffusion model, which only has to train a score model. Training the energy model is around 2x the number of flops as the score model for 3x the cost. However, since training the energy model has substantial benefits performance-wise over only training a score model, this additional cost is a necessary price to pay.
>
> > The proposed method appears to be reminiscent of annealed Langevin dynamics and predictor-corrector samplers discussed in [9][...]. Could the authors discuss the relationship and trade-offs? What are the advantages of your approach?
>
> First, we would like to address the potential confusion about the usage of the term "annealing". Although it is sometimes used for different noise scales of the target distribution (as in [9]), in this paper, we use annealing exclusively for temperature annealing (i.e., changing the power of the marginal density $p_t(x)^\beta$). Note that changing the temperature $p_t(x)^\beta$ and adding the noise $p_t(x) = \int dy\; \mathcal{N}(x|y,\sigma_t^2) * p_0(y)$ are completely different operations resulting in different marginals and requiring different models for their simulation. We will add the corresponding discussion to the text to clarify this terminology.
>
> Second, the Predictor-Corrector scheme proposed in [9] samples from the correct marginals only in the limit of an infinite number of corrector steps (Langevin dynamics), while our proposed inference-time annealing uses a consistent SMC estimator in finite steps by incorporating the importance weights.
>
> > In [10], the authors proposed an energy model with adaptive temperature control by reparameterizing the energy function [...]. Could the authors comment on this approach and discuss the advantages of PITA compared to such adaptive-temperature energy models?
>
> We thank the reviewer for their question. The work in question [10] does not consider sampling from the temperature annealed marginals. Instead, at inference, it runs samples from the marginals with different noise levels, which is similar to the Predictor-Corrector scheme, and cannot be used directly for consistent sampling.
>
> ## Concluding remarks
> We thank the reviewer again for their feedback. We hope that our response and additional experiments clarify all the great points raised in the review and help the reviewer understand the core novelty of our contributions and caliber of our empirical evaluation better. If the reviewer is satisfied with all of our responses, we very politely encourage the reviewer to potentially upgrade their score if they deem it appropriate. We are also more than happy to answer any follow-up questions the reviewer may have.
>
> ## Additional References
>
> [11] Skreta, Marta, et al. "Feynman-kac correctors in diffusion: Annealing, guidance, and product of experts." ICML (2025).

---

> > ### Author Response · Authors · 2025-08-07
> >
> > We thank the reviewer for their time invested in considering our work. We hope our previous response, including the clarifications around novelty and additional experiments, has strengthened the work for the reviewer and addressed their concerns. As the discussion period is coming to a close, we politely remind the reviewer that we remain available to address any final questions they may have.

---

### Note · Authors · 2025-08-12

Dear SAC and AC,

We would like to express our sincere gratitude to all reviewers for their constructive engagement. We are encouraged that our clarifications and new experiments have resolved initial concerns and led to improved scores when warranted. Below, we summarize how we addressed each reviewer’s comments, leading to a consensus in favour of our work.

- **Reviewer JqJ5 (Score increased 3→4):** The reviewer initially raised questions about PITA's core novelty and scope of evaluation. We provided a detailed breakdown of PITA's simultaneous diffusion–temperature annealing novelty and added comparisons with classical baselines (Table R1). The reviewer raised their score, stating, “Thank you for your comprehensive rebuttal…You have addressed my concerns.”
- **Reviewer Ldmh (Score increased 4→5):** Following questions about amortization benefits, we included SMC and PT baselines and wall-clock times, showing better mode coverage of PITA given an equivalent number of energy evaluations (Table R1). We also included experiments on diffusion-only baselines (Table R2). The reviewer was convinced by our rebuttal, stating, “I would like to thank the authors for their detailed reply and the new results… I will raise my score.”
- **Reviewer 9xad (Maintained Score 4):** We engaged in detailed technical discussions, clarifying baseline scalability and the need for EBM distillation for consistent importance sampling, and committed to adding related work and plots. We also clarified limitations of SDE-drift–only estimators, provided a clear inference algorithm, and reported AIS-only ESS values. We believe this addressed all of the reviewer’s concerns.
- **Reviewer SiSU (Maintained Score 4):** Following the reviewer’s suggestions, we added SMC (Table R1), IS ablations (Table R2), and a 40-mode GMM experiment (Table R3) to further showcase PITA’s performance. GMM serves as a useful complementary illustration, as PITA’s capabilities have been shown on more complex tasks. We also clarified how PITA is initialized from a single random sample, using high-temperature MCMC/MD to explore multiple modes—an approach applied across all experiments, addressing neural samplers’ exploration challenges.

We believe our new experiments and clarifications have answered all reviewers’ concerns and position PITA as a new benchmark for diffusion-based Boltzmann sampling in Cartesian coordinates. We hope the AC considers the detailed history of this discussion in their final assessment.

---

### Decision · Program_Chairs · 2025-09-17

**Decision:**

Accept (spotlight)

**Comment:**

The paper present a novel contribution based on good theoretical foundation and with solid empirical results.